# Comparative Transcriptome Analysis Reveals Complex Physiological Response and Gene Regulation in Peanut Roots and Leaves under Manganese Toxicity Stress

**DOI:** 10.3390/ijms24021161

**Published:** 2023-01-06

**Authors:** Ying Liu, Min Zhao, Jingye Chen, Shaoxia Yang, Jianping Chen, Yingbin Xue

**Affiliations:** 1Department of Biotechnology, College of Coastal Agricultural Sciences, Guangdong Ocean University, Zhanjiang 524088, China; 2Department of Food Science and Engineering, College of Food Science and Technology, Guangdong Ocean University, Zhanjiang 524088, China; 3Department of Agronomy, College of Coastal Agricultural Sciences, Guangdong Ocean University, Zhanjiang 524088, China

**Keywords:** peanut, transcriptomic analysis, roots, leaves, Mn toxicity stress

## Abstract

Excess Manganese (Mn) is toxic to plants and reduces crop production. Although physiological and molecular pathways may drive plant responses to Mn toxicity, few studies have evaluated Mn tolerance capacity in roots and leaves. As a result, the processes behind Mn tolerance in various plant tissue or organ are unclear. The reactivity of peanut (*Arachis hypogaea*) to Mn toxicity stress was examined in this study. Mn oxidation spots developed on peanut leaves, and the root growth was inhibited under Mn toxicity stress. The physiological results revealed that under Mn toxicity stress, the activities of antioxidases and the content of proline in roots and leaves were greatly elevated, whereas the content of soluble protein decreased. In addition, manganese and iron ion content in roots and leaves increased significantly, but magnesium ion content decreased drastically. The differentially expressed genes (DEGs) in peanut roots and leaves in response to Mn toxicity were subsequently identified using genome-wide transcriptome analysis. Transcriptomic profiling results showed that 731 and 4589 DEGs were discovered individually in roots and leaves, respectively. Furthermore, only 310 DEGs were frequently adjusted and controlled in peanut roots and leaves, indicating peanut roots and leaves exhibited various toxicity responses to Mn. The results of qRT-PCR suggested that the gene expression of many DEGs in roots and leaves was inconsistent, indicating a more complex regulation of DEGs. Therefore, different regulatory mechanisms are present in peanut roots and leaves in response to Mn toxicity stress. The findings of this study can serve as a starting point for further research into the molecular mechanism of important functional genes in peanut roots and leaves that regulate peanut tolerance to Mn poisoning.

## 1. Introduction

Manganese (Mn) is the active center ion in at least 35 kinds of enzymes, including catalase (CAT) and Mn superoxide dismutase (Mn-SOD) [1,2]. They perform critical functions in plant photosynthesis, respiration, protein synthesis, and hormone activation [1]. Although Mn is an essential mineral for plant growth and development, excessive Mn may be hazardous to plants [3]. Plants only require 20–40 mg/kg Mn (dry weight) to sustain normal growth and development, whereas the Mn concentration in most plants is typically 30–500 mg/kg (dry weight) [4]. Therefore, the Mn level in many plants is above the typical requirement, and excess Mn may impair plant development [5]. Mn toxicity has become a major problem in many regions of the world, thus restricting crop development and decreasing agricultural output in acidic soil [3,4].

Plants have suffered from Mn toxicity stress in recent years as a result of soil acidification and Mn pollution, and the Mn content stored in most plants has considerably surpassed their physiological needs [6]. Excessive Mn buildup in plants may have harmful effects. When plants are subjected to Mn poisoning, visible signs appear on the plant leaves. Plants with brown Mn oxidation patches on their leaves include *Stylosanthes guianensis*, barley (*Hordeum vulgare*), and *Vigna unguiculata* [7,8]. Mn poisoning slows plant development, decreases the number of lateral roots, and reduces root vigor. It also blocks the activities of several enzymes and affects chlorophyll synthesis, resulting in a reduction in the amount of chlorophyll and the effectiveness of photosynthesis in plants [9]. It affects the production of numerous hormones [10]. When crops are subjected to Mn toxicity, the concentration of indoleacetic acid in the body decreases, thus inhibiting leaf growth and stomatal opening. This process subsequently affects the CO_2_ assimilation response in crop photosynthesis, resulting in a considerable loss in crop production and quality [11,12]. Furthermore, excessive Mn deposition in food crops might impair the health of humans via the food chain [13] through conditions such as Parkinson’s syndrome. It can also impair the regular functioning of the digestive, cardiovascular, immunological, and reproductive systems [14]. Consequently, the harm caused by Mn toxicity stress to crops is very direct and severe, thus affecting crop growth and development.

Plants have developed several adaptive methods against Mn toxicity, including absorption control, differential isolation of Mn in subcellular structures, activation of the antioxidant enzyme system, and increased production and secretion of acidic organic compounds to change Mn to the inactive state [1,2,8,15]. Mn toxicity, for example, enhances the activity of peroxidase to promote Mn compartmentation in the apoplast of cowpea (*Vigna sinensis*) and induces excess Mn oxidation [7,16]. Sequestering excess Mn into vacuoles may play a vital function in plant responses to Mn poisoning in leaves [17]. Furthermore, numerous Mn transporters that transport Mn into vacuoles have been identified, including *Oryza sativa* OsMTP8.1, *Arabidopsis thaliana* AtMTP11, and *Cucumis sativus* CsMTP8, indicating that plant Mn tolerance may be mediated by Mn transporters [17,18,19]. Increased organic acid exudation in roots improves tolerance to excess Mn via chelating excessive Mn [8,20]. Furthermore, increased oxalate and citrate root exudates in ryegrass (*Lolium multiflorum*) can limit Mn absorption, thus increasing tolerance to Mn toxicity stress [20]. And yet enhanced malate secretions have a significant impact on the tolerance of Mn for *Stylosanthes guianensis* [8]. Furthermore, roots can improve tolerance to Mn toxicity stress by controlling the uptake of mineral nutrients, including magnesium (Mg), calcium (Ca), and iron (Fe) [21,22,23].

Peanut (*Arachis hypogaea* L.) is a significant cash crop and oil crop that offers edible oil and protein to people all over the world [24,25]. Peanuts, as agricultural crops, frequently confront numerous metal stressors during their life cycle, thus restricting productivity and endangering human health owing to hazardous metal buildup [26,27]. Peanuts are extremely vulnerable to Mn poisoning. Peanut seedling development is impeded when the concentration of soluble (Mn^2+^) exceeds 150 μM. [28]. Mn stress can impair plant ion absorption and transport, resulting in visible Mn oxidation spots on leaves, which can lead to reduced chlorophyll synthesis, decreased photosynthetic rate, reactive oxygen species buildup, and disturbance of hormone balance in leaves [9,10]. Therefore, Mn toxicity is among the major variables that influence peanut growth and limit peanut yield. A method for enhancing peanut tolerance against Mn toxicity stress needs to be determined.

In recent years, high-throughput transcriptome sequencing technology has been extensively employed to study the response mechanisms of plants to toxic stresses of heavy metals, such as copper, lead, aluminum, and cadmium, including *Citrus grandis*, *Raphanus sativus*, *O. sativa*, and *Saccharum officinarum* [29,30,31,32]. However, limited studies have focused on manganese-induced stress in peanut plants, and the molecular regulatory mechanism of peanut plants in response to Mn stress remains unknown. Mn toxicity response genes in roots and leaves were compared using high-throughput deep sequencing technologies. Furthermore, the physiological parameters of the appearance of Mn oxidation spots and the concentration of metal ions (Mn, Mg, and Fe) in roots and leaves were both studied at different concentrations of Mn treatment conditions. Our findings will reveal more information on the unique molecular regulatory pathways that underpin root and leaf response to Mn toxicity. The results of this study may provide a preliminary basis for additional research into the specific functions of genes that are sensitive to Mn toxicity.

## 2. Results

### 2.1. Mn Toxicity Stress on Peanut Growth and Development

Two concentrations of normal Mn (10 µM) and Mn toxicity (300 µM) were used to study the effects of Mn toxicity stress on peanut development. Mn poisoning impeded peanut development, as evidenced by the lower peanut plant height, SPAD, fresh weight, and dry weight and the growing Mn poison spots on the leaves (Figure 1 and Table 1). Plant height dropped by 16.61% when Mn concentrations increased from 10 µM to 300 µM. (Figure 1A,B and Table 1). Furthermore, the SPAD value, which reflects the chlorophyll content, declined by 10.71%. (Table 1). Likewise, the fresh weight of shoots, fresh weight of roots, dry weight of shoots, and dry weight of roots were reduced by 63.89%, 56.69%, 54.04%, and 52.80%, respectively (Table 1). Furthermore, brown spots, which indicate Mn poisoning, were found when Mn concentrations in leaves were 300 μM, and the density of brown spots accrued with raising the concentration of Mn; evidence indicated the following a 60-times rise at the concentration of 300 μM Mn in comparison with that at 10 μM Mn (Figure 1C,D and Table 1).

### 2.2. Influences of Mn Poisoning Stress on the Growth of Peanut Root

Mn toxicity stress influenced the growth of peanut root as exogenous Mn concentration increased (Table 2). The volume of roots, surface area of roots, total root length, and the number of root tips remarkably decreased by 90.97%, 66.98%, 58.93%, and 49.38%, respectively, as the additive Mn concentration was at 300 μM and the cultivation duration was 20 days compared with the control Mn (10 μM, Table 2). Nonetheless, no substantial change was observed in the root diameter (Table 2).

### 2.3. Influences of Mn Poisoning Stress on Peanut Physiological Response Indicators

Many peanut physiological indices were influenced by Mn toxicity stress, and the responses of each physiological index in root and leaf to Mn toxicity varied (Table 3). When peanut was subjected to Mn toxicity stress (300 μM of MnSO_4_), the activities of POD, APX, and SOD in leaves remarkably increased by 800.00%, 191.93%, and 53.42%, respectively (Table 3), whereas the activities of CAT in leaves remarkably decreased by 14.95% (Table 3). In contrast, the amount of soluble protein and MDA in leaves dramatically dropped by 7.56% and 21.21%, whereas the amount of proline in leaves substantially increased by 3.16 times (Table 3). In parallel, APX and SOD activity in the root considerably increased by 82.26% and 21.63%, respectively (Table 3). POD activity did not change much, while the activity of CAT in the root fell dramatically by 73.33% (Table 3). Proline and soluble protein concentrations in the root rose considerably by 13.94% and 104.68%, respectively, although MDA concentrations did not change remarkably (Table 3).

### 2.4. Results of Mn Effectiveness on Three Metal Ions Concentrations in Peanut Leaves and Roots

The concentrations of three metal ions (Mn, Mg, and Fe) in peanut leaves and roots were thoroughly investigated at various levels of Mn (Figure 2). The concentration of Mn rose with increased Mn availability, as evidenced by a 4.07- and 8.17-fold increase at 300 μM Mn compared with 10 μM Mn, respectively (Figure 2A). Unlike the varying in the concentration of Mn in peanut roots and plants in reaction to Mn poisoning, Mg concentration in roots and leaves declined (Figure 2B). The Mg content in roots and leaves fell by 32.40% and 36.08%, respectively, at the concentration of 300 μM Mn treatment groups in comparison with the 10 μM Mn treatment groups (Figure 2B). However, Fe concentration remarkably increased under 300 μM Mn treating, as demonstrated by 44.81% and 37.93% higher concentrations than that under 10 μM Mn treatment (Figure 2C).

### 2.5. Peanut Root and Leaf Transcriptome Profiling in Response to Mn Poisoning

Whole genome transcriptome sequencing analysis was applied to study the transcriptomics of peanut leaves and roots under 10 and 300 μM Mn treatments to investigate the molecular responses of peanut leaves and roots to Mn poisoning stress. In the two Mn treatments, 12 libraries were built for the roots and leaves. These libraries generated roughly 36.11–55.27 million raw reads and approximately 33.60–51.43 million clean reads (Appendix A). In total, 52,320 and 47,675 genes were discovered in the roots and leaves, respectively (Table 4). In the roots and leaves, 749 and 4589 genes were differently expressed at the two Mn levels (Table 4 and Appendix A).

A total of 310 DEGs were discovered in both roots and leaves, comprising 272 upregulated and 23 downregulated DEGs. Furthermore, five DEGs were upregulated in roots in reaction to Mn poisoning while downregulated in leaves (Appendix A). Ten DEGs were downregulated in the roots in response to Mn toxicity but were upregulated in the leaves (Table 4 and Appendix A). Therefore, identical reactions happened in both leaves and roots of peanut plants suffering from Mn poisoning.

All of the 4718 DEGs revealed differently controlled models by Mn poisoning in roots compared with the leaves (Table 4 and Appendix A), indicating the distinct reactions between roots and leaves under Mn toxicity. In total, 265 and 3284 upregulated DEGs were detected in the leaves and roots, respectively, whereas 174 downregulated DEGs were found in peanut roots, and 995 DEGs were observed in peanut leaves (Appendix A).

### 2.6. Analysis of Functional Enrichment of DEGs

According to MF (molecular function), BP (biological process), and CC (cellular component), the result analysis of GO enrichment of DEGs in peanut roots and leaves was categorized. The maximum 10 GO term entries with the minimum *p*-value, in which the most significant enrichment was observed, were selected for each GO classification and presented. The biological process BP contains the majority of the DEGs in leaves and roots (Appendix A).

The Rich factor, the value of FDR, and the quantity of genes enriched to this GO term were used to calculate the degree of enrichment based on the GO enrichment data. The rich factor is the rate of the quantity of DEGs enriched to the quantity of DEGs annotated in the GO term, in which a higher rich factor indicates greater enrichment, and a larger FDR indicates greater significance. The enrichment in leaves was mainly observed in the plasma membrane (a total of 151 genes, containing 120 up-regulation and 31 down-regulation genes), carbohydrate metabolic process (a total of 195 genes, containing 164 up-regulation and 31 down-regulation genes), transferase activity (a total of 611 genes, containing 500 up-regulation and 111 down-regulation genes), oxidoreductase activity (a total of 348 genes, containing 273 up-regulation and 75 down-regulation genes), catalytic activity (1400 genes, including 1122 upregulated and 278 downregulated genes), integral components of membranes (a total of 782 genes, containing 611 up-regulation and 171 down-regulation genes), components of membranes (a total of 795 genes, containing 624 up-regulation and 171 down-regulation genes), and membranes (a total of 1001 genes, containing 780 up-regulation and 221 down-regulation genes) in leaves (Figure 3A and Appendix A). In the roots, enrichment was mainly observed in ion homeostasis (a total of 16 genes, containing 11 up-regulation and five down-regulation genes), inorganic ion homeostasis (six downregulated genes), cation homeostasis (a total of 16 genes, containing five up-regulation and 11 down-regulation genes), cellular ion homeostasis (a total of 15 genes, containing four up-regulation and 11 down-regulation genes), cellular cation homeostasis (a total of 15 genes, containing four up-regulation and 11 down-regulation genes), metal ion homeostasis (a total of 15 genes, containing four up-regulation and 11 down-regulation genes), cellular metal ion homeostasis (a total of 15 genes, containing four up-regulation and 11 down-regulation genes), and cellular transition metal ion homeostasis (a total of 15 genes, containing four up-regulation and 11 down-regulation genes) in roots (Figure 3B and Appendix A).

### 2.7. Analysis of DEGs That Act as Conduits and Transporters

In all of the 97 DEGs that function as conduits and transporters were authenticated, including seven DEGs coding for *oligopeptide transporter*, 19 genes coding for *calcium-transporting ATPase*, six genes coding for *metal-nicotianamine transporter* (*YSL*), five genes coding for *high-affinity nitrate transporter*, four genes coding for *metal tolerance protein* (*MTP*), 12 genes coding for *potassium transporter*, nine genes coding for *sulfate transporter*, nine genes coding for *aluminum-activated malate transporter*, 11 genes coding for *vacuolar iron transporter*, six genes coding for *zinc transporter*, two genes coding for *boron transporter*, four genes coding for *magnesium transporter*, and three genes coding for *calcium-permeable stress-gated cation channel* in roots or leaves (Table 5). A total of two out of seven *oligopeptide transporters*, one out of six *YSLs*, one out of five *high-affinity nitrate transporters*, two out of four *MTPs*, three out of 12 *potassium transporters*, three out of nine *sulfate transporters*, two out of nine *aluminum-activated malate transporters,* eight out of 11 *vacuolar iron transporters*, one out of six *zinc transporters*, and one out of two *boron transporters* were authenticated just in roots (Table 5). Nevertheless, the DEGs of those families demonstrated diverse expression models in roots; for instance, two up-regulation and one down-regulation DEGs were found for *oligopeptide transporters*, one up-regulation and one down-regulation DEG was found for *high-affinity nitrate transporters*, four up-regulation and one down-regulation DEGs were found for *potassium transporters*, one up-regulation and three down-regulation DEGs were found for *sulfate transporters*, two up-regulation and one down-regulation DEGs were found for *aluminum-activated malate transporters*, and one up-regulation and one down-regulation DEGs were found for *zinc transporters* (Table 5). Interestingly, one *oligopeptide transporter* (i.e., AH14G01590), six *calcium-transporting ATPases* (i.e., AH14G23400, AH20G34940, AH06G11590, AH16G01740, AH16G01740, AH01G07000), one *YSL* (i.e., AH15G32580), one *high-affinity nitrate transporter* (i.e., AH03G40290), two *potassium transporters* (i.e., AH16G06470, AH06G03660), one *sulfate transporter* (i.e., AH20G08820), one *aluminum-activated malate transporter* (i.e., AH05G32880), two *vacuolar iron transporters* (i.e., AH05G36140, AH15G37400), one *zinc transporter* (i.e., AH05G30820), and one *boron transporter* (i.e., AH08G26430) were identified simultaneously in peanut leaves and roots (Table 5). One *oligopeptide transporter*, one *YSL,* one *sulfate transporter*, and one *boron transporter* were up-regulated in peanut leaves while down-regulated in peanut roots. Conversely, *zinc transporter* was downregulated in peanut leaves and upregulated in peanut roots in reaction to Mn poisoning (Table 5).

### 2.8. Distinguishing of DEGs That Act as Antioxidant Substances

All of the 57 DEGs playing a role in antioxidation were distinguished as having different reactions to Mn poisoning stress in leaves and roots (Table 6). Of all the DEGs, 18 DEGs were found just in the roots, namely, the 18 *PODs*, as reflected by 10 upregulated and 8 downregulated genes in reaction to Mn poisoning (Table 6). In total, 34 DEGs were found just in peanut leaves, including 3 *L-ascorbate oxidases*, one *SOD*, and 30 *PODs* (Table 6). The three *L-ascorbate oxidases* (AH13G38160, AH17G04410 and AH03G34360) and one *SOD* (AH19G19840) showed the same expression patterns, and these genes were upregulated only in leaves in reaction to Mn poisoning (Table 6). Nevertheless, different modes of expression were found amongst the 30 *PODs*, as demonstrated by the 24 upregulated and six downregulated ones only in peanut leaves in reaction to Mn poisoning (Table 6). It is interesting that five *PODs* (i.e., AH18G05400, AH09G31660, AH04G09840, AH14G08440, and AH14G25410) were identified simultaneously in peanut leaves and roots (Table 6), and one of them (i.e., AH18G05400) was upregulated in the roots but downregulated in peanut leaves in response to Mn poisoning (Table 6). However, two genes, namely, AH04G09840 and AH14G08440, were down-regulated in peanut roots while up-regulated in leaves in reaction to Mn poisoning (Table 6). Moreover, AH09G31660 was downregulated simultaneously in peanut leaves and roots, while AH14G25410 was upregulated concurrently in peanut leaves and roots in reaction to Mn poisoning (Table 6).

### 2.9. Determination of DEGs That Serve as Transcription Factors

All of 147 DEGs playing a key role in transcription factors were found, including 30 DEGs encoding for *WRKY transcription factors* (*WRKYs*), 39 DEGs coding for *ethylene-responsive factors* (*ERFs*), 21 DEGs coding for *Myb family transcription factors* (*MYBs*), 16 DEGs coding for *bHLH transcription factors* (*bHLHs*), two DEGs coding for *AP2/ERF and B3 domain-containing transcription factors* (*AP2/ERFs*), two DEGs coding for *bZIP transcription factors* (*bZIPs*), four DEGs coding for *GATA transcription factors* (*GATAs*), eight DEGs coding for *heat stress transcription factors* (*HSTFs*), one DEG coding for *MADS-box transcription factor* (*MADS*), one DEG coding for *NAC transcription factor* (*NAC*), one DEG coding for *nuclear transcription factor* (*NTF*), two DEGs coding for *BEE transcription factor* (*BEEs*), two DEGs coding for *CPC transcription factor* (*CPCs*), two DEGs coding for *DIVARICATA transcription factor* (*DIVARICATAs*), one DEG coding for *FAMA transcription factor* (*FAMA*), three DEGs coding for *HBP-1b transcription factor* (*HBP-1bs*), one DEG coding for *KAN transcription factor* (*KAN*), one DEG coding for *ORG transcription factor* (*ORG*), one DEG coding for *PERIANTHIA transcription factor* (*PERIANTHIA*), three DEGs coding for *TCP transcription factor* (*TCPs*), one DEG coding for *TGA transcription factor* (*TGA*), two DEGs coding for *UNE transcription factor* (*UNEs*), and three DEGs coding for *trihelix transcription factors* (*TTFs*) in roots or leaves (Table 7). Three out of 16 *bHLHs*, two out of 39 *ERFs*, one *KAN*, three out of 21 *MYBs*, one out of three *TCPs*, one out of three *TTFs*, and one out of 30 *WRKYs* were found only in roots (Table 7). In contrast, the DEGs of those gene families demonstrated diverse modes of expression in roots. For instance, one up-regulation and one down-regulation DEG were known as *ERFs*, and one up-regulation and two down-regulation DEGs were identified as *MYBs* (Table 7). Interestingly, seven other *WRKYs* (AH06G25830, AH08G09100, AH13G32420, AH03G28760, AH08G28680, AH16G13340, and AH12G03520) and two *ERFs* (AH16G37720 and AH20G26610) were identified in both roots and leaves (Table 7). All seven *WRKYs* and two *ERFs* were up-regulated synchronously in peanut leaves and roots (Table 7).

### 2.10. qRT-PCR Analysis of DEGs

The results of transcriptome sequencing were confirmed, and qRT-PCR testing of 19 DEGs was implemented by harvested peanut leaves and roots from the experimental group (300 μM Mn) and the control group (10 μM Mn; Figure 4). The genes for testing contained seven genes that worked in transport, three genes that took effect in responses to stress, and nine genes that functioned as transcription factors (Figure 4). 

According to qRT-PCR analysis, 18 genes of the evaluated genetic transcription were up- or down-regulated in response to Mn poisoning in the roots. Mn poisoning increased the transcription of five genes that served as transport, including *yellow streak protein* (*YSL1a/2*), *metal resistance protein* (*MTP10.2*), *aluminum-activated malate transporter protein* (*ALMT*), and *magnesium transporter protein* (*MAT*). Two genes were involved in stress, including *APX* and *SOD protein*. Nine genes functioned as transcription factors, including *ethylene response transcription factor* (*ERF1/2*), *transcription factor of heat stress* (*HST1/2*), *transcription factor of MYB* (*MYB*), *transcription factor of bHLH* (*bHLH1/2*), and *WRKY transcription factor* (*WRKY1/2*, Figure 4A). These findings support the results of the transcriptome sequencing (Appendix A). For comparison, the gene transcription of *YSL3a*, *MTP10.1,* and *peroxidase* (*POD*) was restrained by Mn poisoning (Figure 4A), supporting the results of transcriptome sequencing (Appendix A).

Mn poisoning influenced the expression levels of all 19 examined genes in the leaves (Figure 4B). Promoted gene transcription was found in five examined genes that participated in transporting, including *YSL1a/2/3a*, *ALMT* and *MAT*. Three examined genes participated in responses to stress, including *APX, POD* and *SOD*, and four genes involved in transcription factors, including *ERF1*, *HST1/2,* and *bHLH* (Figure 4B). By contrast, the expression levels of *MTP10.1/10.2*, *ERF2*, *MYB,* and *WRKY1/2* were downregulated in peanut leaves in responding to Mn poisoning (Figure 4B). Those findings agreed with the testing results of transcriptome sequencing (Appendix A). 

When peanut was subjected to Mn stress, quantitative results showed that different functional DEGs demonstrated diverse modes of expression in peanut leaves and roots, suggesting their different functional roles in roots and leaves, which may indicate the existence of different complex regulatory mechanisms for different DEGs (Figure 5). Some DEGs may result in changes in physiological indicators, and the physiological results showed that POD, APX, SOD, and proline were remarkably upregulated in peanut roots and leaves, whereas CAT was substantially downregulated (Figure 5). This condition might lead to a significant decrease in peanut plant height and phenotypes, such as Mn spots on leaves. Mn spots on leaves might lead to a decrease in chlorophyll content, which might subsequently affect ion transport in the plant. The ion content measurements showed a significant increase in the content of Mn and Fe and a significant decreasing in the content of Mg in peanut leaves and roots (Figure 5). Therefore, peanut roots and leaves might have complex regulatory mechanisms in response to Mn toxicity stress, thus requiring further in-depth studies.

## 3. Discussion

Excess of available Mn is hazardous to crops and a constraint for agricultural development, particularly in acid-soil [1,14,33]. Generally, various types of crops have varying levels of toleration to Mn poisoning. For instance, soybean is more susceptible to Mn poisoning than *Stylosanthes guianensis* [8,33,34]. Nevertheless, the distinct response at the molecular level of different parts of crops to Mn poisoning remained unknown. In the present study, the fresh weight of peanut shoots and roots remarkably reduced while the concentration of Mn enhanced to 300 μM. Excess Mn has an inhibitory effect on cucumber (*Cucumis sativus*) growth, resulting in a significant reduction in above- and below-ground dry weight [35]. Under Mn toxicity stress, the biomasses of the above-ground part and underground part of soybean are severely decreased, and the development of roots is impeded in some ways [14]. This study showed the aboveground and root biomass were also decreased observably, and the growth of roots was restrained to some degree when the peanut plant was subjected to Mn toxic stress. This phenomenon occurred, possibly because the excessive Mn accumulation of plants may result in severe injury of cells [1,36], ultimately influencing the normal growth and development of the plant.

In the present study, Mn toxicity caused Mn spotting and wrinkling of peanut leaves, which correlates with previous related reports in *O. sativa*, *Stylosanthes guianensis*, and soybean [14,37,38]. The generation of Mn oxide spots is primarily attributed to the increasing amount of Mn compounds or oxidized phenol-like substances in the outside cell wall of the leaf epidermis of these plants under Mn-toxic conditions [1,39]. In this study, under Mn toxicity stress, peanut leaves showed obvious Mn oxidation spots, indicating that peanut leaves also accumulate large amounts of Mn oxides or oxidized phenol-like substances.

Mn can aid in the maintenance of the chloroplast membrane’s normal shape by taking part in the systems of photosynthesis electron transfer and photolysis of water [39]. Fe is participated in the process of photosynthesis and the electron transfer system in respiration and influences the development of chloroplast, which is required for the formation of chlorophyll [2]. Mg is required for the production of plant chlorophyll and is vital for the metabolism of plants [2,21]. As a result, the balance and stability of the relative levels of Mn, Mg and Fe might be critical for chlorophyll production and the process of photosynthesis. In this study, when peanuts suffered from Mn poisoning stress, although the accumulation content of Mn in both peanut roots and leaves was higher, the Mn content in the roots was notably more than that in peanut leaves, and the transfer or enrichment of superabundant Mn in the roots might be a crucial mechanism for peanut to mitigate its poisoning effect. Therefore, Mn might have some distribution mechanism in the roots and leaves, but this molecular mechanism has not been fully understood.

Unlike the varied pattern of Mn content in plants, Mg content in peanut roots and leaves declined with increasing exogenous Mn concentration, demonstrating that Mn and Mg absorption had antagonistic effects. The absorption of Mg in *S. guianensis*, *Solanum lycopersicum*, *Sorghum bicolor,* and other plants was blocked, and the Mg concentration in plants was dramatically lowered, similar to earlier study results [40]. In this study, Mn toxicity stress considerably lowered Mg concentration in peanut roots and leaves, indicating that Mn toxicosis stress primarily impeded Mg absorption by peanut roots and leaves. Reduced Mg concentrations in the roots and leaves might play an important role in reducing Mn poisoning effectiveness and preserving normal root and leaf function, which might be one of the accommodation processes for peanuts suffering from Mn poisoning.

Furthermore, boosting Fe concentrations and improving Fe absorption will be beneficial to plants to accommodate responses to Mn poisoning [1]. Fe concentrations in cotton varieties of Mn-resistant strains were greater than those of cotton varieties of Mn-intolerant strains [41]. When peanuts were subjected to Mn poisoning, the contents of Fe in roots and leaves changed distinctly and maintained high concentrations. Therefore, peanut roots and leaves could keep a higher content of Fe to relieve the effect of Mn poisoning, which may serve as a physiological mechanism for peanut plants to adapt to Mn poisoning.

As a normal secondary metabolite in plant cell metabolism, ROS (reactive oxygen species) has a beneficial effect on plant response to environmental stress, depending mainly on whether the delicate balance between ROS production and bursting is disrupted [42]. SOD, APX, and POD are the main antioxidants responsible for eliminating ROS in plants [43,44]. Stress tolerance in plants can be improved by enhancing the vitality and content of antioxidases to reduce the accumulation of ROS in cells [43,44]. In the present study, the activities of POD, SOD, and APX in peanut roots and leaves remarkably increased in responding to Mn poisoning. Therefore, the antioxidant defense system of peanut plants was activated in response to Mn poisoning, and the activities of different enzymes in different parts of the plant were remarkably different. POD primarily took charge of eliminating the oxygen free radicals in peanut leaves, while SOD and APX were mainly responsible for the scavenging of reactive oxygen radicals in peanut roots and leaves. Furthermore, the MDA content is a crucial referent for the level of lipid peroxidation of the membrane in plant cells in a comprehensive manner [45]. In this study, the degree of membrane lipid peroxidation in stressed peanut leaves was less than that in the experimental control group under the joint protection of various protective enzymes, while no significant difference was observed in the roots. High Mn stress may have broken the metabolic balance of intercellular ROS, causing differences in the MDA content in different parts.

Proline has a vital function in osmoregulation, maintenance of plant cell strength, and maintenance of osmotic pressure in the cytoplasm, contributing to the stability of cellular proteins and membranes [46]. Proline also acts as a scavenger of ROS and works synergistically with antioxidant enzymes to reduce ROS in plants [47]. In this study, the contents of proline significantly elevated in both leaves and roots of peanut plants suffering from Mn poisoning, indicating positive regulation in responding to Mn poisoning, which decreased the osmotic pressure of cells and required more proline to maintain osmotic pressure. Soluble proteins can participate in osmoregulation as osmotic regulators and reflect the degree of damage to plant organs [48]. Plant biosynthesis of soluble proteins is affected by abiotic stress [49]. In our study, soluble proteins showed an obvious increase in peanut roots and a significant decrease in leaves, which could be a response of soluble proteins to Mn stress. In addition, considering that soluble protein degradation produces a large amount of free amino acids, proline is one of the first rapidly increasing amino acids in various crops, and the increase in proline content may be related to soluble protein breakdown [50].

Although genome-wide identification of DEGs in different types of plants in response to stressors of heavy metal ions has been examined, only one study in grape (*Vitis vinifera*) roots in terms of Mn toxicity has been published [51,52,53,54]. A total of 2629 and 3278 DEGs were discovered in response to Mn poisoning in grape roots of two distinct varieties, namely, Combier and Jinshou, separately, indicating that Combier and Jinshou have differing tolerances to Mn toxicity [54]. Limited genome-wide investigations have focused on leaf responses to Mn toxicity [2,55], but a comparison analysis of roots and leaves under Mn toxicity stress has not been carried out. In the present work, RNA-seq was used to conduct a whole genome investigation of the responding of DEGs to Mn poisoning stress on peanut roots and leaves, and 749 and 4589 DEGs were discovered from the peanut roots and leaves separately. Therefore, peanut roots and leaves might have very distinct molecular pathways.

Plant respiratory action, oxygenic photosynthesis, and activities of various secondary metabolism are all influenced by Mn poisoning [7,56]. In this study, a significant number of DEGs were functionally enriched involving MF, BP and CC in peanut leaves and roots, demonstrating that complicated metabolic alterations might exist in peanut leaves and roots in responding to Mn poisoning. 

Excess Mn translocation is a mechanism employed by many plants to adjust to Mn poisoning, and this process is primarily adjusted and controlled by metal-ion transporter [23,57]. For instance, genetically modified rice with down-regulated expression of *a metal-nicotianamine transporter* named *YSL* (*yellow stripe-like member*) showed a substantial drop in Mn content in rice grains, suggesting that it acts in managing the long-distance transmission of Mn-nicotianamine in plants [57]. In our study, five *metal-nicotianamine transporters* were upregulated in peanut leaves in response to Mn poisoning, intensely indicating that transporters of metal-nicotianamine might participate in Mn transporting in peanut leaves in Mn poisoning circumstances. The activities of *oligopeptide transporters* are comparable to those of Mn recombination transporters in *Arabidopsis thaliana* [1,58]. In the present work, seven *oligopeptide transporters* were found in peanut roots and leaves in responding to Mn poisoning. Therefore, altering the transshipment of excessive Mn from peanut roots and leaves via changed transcriptions of *metal-nicotianamine transporters* and *oligopeptide transporters* might be essential for peanut tolerance to Mn poisoning.

Mn subcellular compartmentalization has an important effect on plants’ resistance to Mn poisoning [36]. AtECA1 and AtECA3 are localized respectively in the endoplasmic reticulum and Golgi body, which are both belonged to calcium ATPases, direct adjusting excessive Mn transporting into the endoplasmic reticulum and Golgi body in *A. thaliana*, separately [59,60]. Under Mn toxicity circumstances, the mutations of *AtECA1* or *AtECA3* can impede root development and cause serious chlorosis on the leaves of *A. thaliana* [59,60]. In the present work, six *calcium-transporting ATPases* were upregulated in both peanut leaves and roots in responding to Mn poisoning, indicating that the family genes of *Ca^2+^-ATPase* might participate in Mn detoxifying in peanut roots and leaves. Mn toxicity upregulated two *vacuolar iron transporters* (AH05G36140 and AH15G37400) in both roots and leaves, which were anticipated to be situated in the Golgi, indicating that it might participate in regulating the delineation and compartmentalization of excessive Mn transferring into the Golgi body.

Moreover, members of the family of MTP are important Mn transporters that govern Mn uptake and transshipment in plants [61,62]. CsMTP8.2 operates as an Mn-specific transporter in the tea plant (*Camellia sinensis*) that contributes to the outflow of excess Mn^2+^ from plant cells [62]. OsMTP11 takes part in Mn remobilization in the cell cytoplasm and vacuolar membrane and may have an important effect on the transshipment of Mn and other heavy metals in *O. sativa* [61]. In this study, four *AhMTPs* were discovered in response to Mn poisoning in roots or leaves in our investigation. Mn toxicity upregulated two *AhMTPs* (AH10G30290 and AH13G56980), AH16G14430, and AH09G00220 in roots and leaves, suggesting that the four *AhMTPs* might participate in peanut acclimatizing to Mn poisoning by adjusting Mn enrichment and isolation. The findings might supply molecular support for the evident Mn poisoning phenotype discovered in peanut root and foliage.

Homeostasis, or the regulation of the uptake and transshipment of other metallic elements such as Mg or Fe, is a key approach for coping with Mn poisoning [2,21]. In the present work, treatment with 300 μM Mn resulted in the decreased expression of a *magnesium transporter* (AH16G35450) and a substantial drop in Mg content in peanut leaves, demonstrating that Mn poisoning might influence the expression of *magnesium transporter* and, consequently, Mg aggregation in peanut leaves. Moreover, an increase in Fe effectiveness alleviates Mn poisoning in *Hypogymnia physodes* [21]. Besides Fe, Mg aids in enhancing wheat (*Triticum aestivum*) tolerance to excessive Mn poisoning [63]. In the present study, Mn poisoning increased the Fe concentrations in both roots and leaves. Furthermore, reduced expression of a *magnesium transporter* (AH16G35450) was detected in peanut leaves exposed to Mn toxicity, suggesting that the gene of *magnesium transporter* might control Mg balancing in peanut leaves in responding to Mn poisoning.

Considering that Mn poisoning may cause an irritable oxidation reaction, controlling the activity of antioxidative enzymes is often regarded as one of the essential Mn poisoning toleration methods [38,64]. In *Pennisetum purpureum*, the expression levels of *PpSOD* are significantly higher in the Mn-tolerance variety with quite high SOD activity, whereas this was not seen in the Mn-intolerance variety, indicating that *SOD* may control tolerance against Mn poisoning stresses in plants [64]. Mn toxicity stress increases POD activity and gene expression in cowpea (*Vigna sinensis*) and stylosanthes (*Stylosanthes guianensis*), respectively, which both have a significant role in adaptation to Mn poisoning for plants [38,39]. In this study, One *SOD* and 24 *PODs* were all upregulated exclusively in the peanut leaves of this research but not in the roots. Therefore, the upregulation expression of one *SOD* gene and 24 *PODs* might be beneficial to the increased Mn toleration of the roots in comparison with the peanut leaves.

Finally, several genes of *transcription factors* were found in this investigation. In brief, 147 *DEGs* of transcription factor were differently sensitive to Mn poisoning stress in peanut leaves and roots. Three of the 16 *bHLH transcription factors* were up-regulated of expression in roots to make the response to Mn poisoning. Already there is evidence that one gene belongs to the family of the *bHLH transcription factor*, namely, *AtNAI1*, influencing *AtMEB1/2* expression levels and governing Mn poisoning toleration in *A. thaliana* [65]. However, the activities of *bHLH transcription factors* family in peanut leaves and roots in response to Mn poisoning were still unclear. Therefore, intricate regulatory systems in peanut roots and leaves did not respond to Mn poisoning.

## 4. Materials and Methods

### 4.1. Plant Material Sources and Processing Methods

The experimental material was the peanut cultivar Zhanyou 62, which was bred by the Zhanjiang Institute of Agricultural Sciences in Guangdong Province, China. Peanut seeds were sprouted into sand for 8 days of culture prior to Mn treatment. Afterward, as previously mentioned, the peanut seedlings with uniform growth were moved to plastic boxes for hydroponics with a volume of 15 L that was added with the nutrient solution [2]. In brief, the nutritional solution contained 1500 μM KNO_3_, 400 μM NH_4_NO_3_, 25 μM MgCl_2_, 1200 μM Ca(NO_3_)_2_·4H_2_O, 40 μM Fe-EDTA (Na), 500 μM MgSO_4_·7H_2_O, 300 μM K_2_SO_4_, 300 μM (NH_4_)_2_SO_4_, 1.5 μM ZnSO_4_·7H_2_O, 0.5 μM CuSO_4_·5H_2_O, 0.16 μM (NH_4_)_5_MoO_24_·4H_2_O, 500 μM KH_2_PO_4_, and 2.5 μM NaB_4_O_7_·10H_2_O. All chemicals used were analytically pure-grade reagents (Kermel, Tianjin, China). Meanwhile, 10 and 300 μM MnSO_4_ (Kermel, China) were individually added to the nutritional solution for Mn treatment. The treatment with an Mn concentration of 10 μM was used as the control group. The experiments were repeated four times for each treatment concentration. Temperatures of 25–30/18–22 °C were used to regulate plant development. The photoperiod lasted for around 12 h/d. The nutritional solution was required to be changed every 5 days. And every 2 days, the pH of the nutrient solution was adjusted to 5.0 with 1 M of KOH or H_2_SO_4_ (Kermel, China). After 20 days of treatment with 10 and 300 μM of Mn, the above-ground and below-ground parts of peanuts were collected individually to determine the fresh and dry weights, the roots and the number of Mn spots on the fifth trifoliate leaves, etc. Furthermore, the amounts of Mn, Mg and Fe in the leaves and roots were determined individually.

### 4.2. Determination of Plant Heights, Fresh and Dry Weights of Peanuts

Immediately after harvesting the peanut plants, the heights of the plant and the fresh weights of the aboveground and subsurface sections were separately measured. Peanut seedlings were placed in an oven (Yiheng, Shanghai, China), drying at a temperature of 105 °C lasting for 30 min, and their dry weights were calculated after drying until the weight no longer changed at the temperature of 75 °C [14]. Each index was repeated 4 times.

### 4.3. Brown Spot Measurement

The 5th trifoliate leaves from the bottom were obtained independently at 20 days under different Mn treatments to calculate Mn toxicity spots in the leaves of peanut plants. The amount of Mn poison spots on the leaves was determined using a previously reported square technique [33]. In brief, the Mn spots in the upper, middle and lower regions of the identical leaf were evaluated using a 1 cm^2^ clear plastic film, and the mean number of Mn spots in 3 squares was determined. Data are expressed as the mean of 4 replicate experiments plus or minus the standard deviation.

### 4.4. Analysis of Morphological Root Parameters

WinRHIZO technology was used to analyze the morphological parameters of peanut seedling roots as previously described [66]. The fresh roots of different treatment groups were collected and completely unfolded onto the platform of the scanner (Epson, Tokyo, Japan). The samples were then evaluated using software for computer image analysis (WinRhizo Pro, Québec, QC, Canada).

### 4.5. Malondialdehyde (MDA) Content Assaying 

The physiological response indicators of peanut roots and leaves were examined after 20 days of dealing with 10 μM (control group, normal Mn concentration) and 300 μM (Mn toxicity stress concentration) MnSO_4_. The content of MDA was determined by using a slightly modified version of the thiobarbituric acid (TBA) technique [67]. In brief, 0.1 g leaf or root tissue was homogenized in 10 mL phosphate buffer (Solarbio, Beijing, China) (pH 7.8, 0.05 M) before being extracted in 2 mL 0.6% TBA (Rhawn, Shanghai, China). The extract was placed in the thermostatic water bath (Shanghai Lichen, China) at 100 °C for 15 min before being rapidly cooled on ice. After spinning at 4000 rpm for 20 min in a centrifugal machine (Eppendorf 5415D, Hamburg, Germany), the absorbance of the liquid supernatant was tested at 450, 532, and 600 nm, respectively, using an ultraviolet-visible spectrophotometer (Shanghai Yuanxi UV-5100B, Shanghai, China) with a 10 mm quartz cuvette (Allrenta, Beijing, China). The compound of MDA-TBA was quantized by the extinction coefficients (155 mM^−1^ cm^−1^).

### 4.6. Soluble Protein Content Assaying 

The coomassie brilliant blue technique was used to determine the amount of soluble protein [68]. 0.1 g leaf or root tissue were homogenized respectively in 10 mL phosphate buffer (pH 7.8, 0.05 M) before being extracted in a 2.9 mL liquid mixture containing 0.1 g Coomassie brilliant blue G-250 (Rhawn Chemistry, Shanghai, China). After a 2-min reaction, the absorbance of the liquid supernatant was tested at 595 nm to calculate the protein concentration in the sample by using a standard curve via bovine serum albumin (BSA; Rhawn Chemistry, China).

### 4.7. Proline Content Analysis 

Leaf and root samples (0.1 g each) were mixed in 10 mL 3% sulfosalicylic acid (Rhawn Chemistry, China) before filtering to evaluate the proline concentrations of the leaves and roots, respectively [69]. The reaction mixture contained 2 mL extracted supernatant, 3 mL acid ninhydrin reagent (Rhawn Chemistry, China), and 2 mL glacial acetic acid (Ghtech, Shantou, China) was placed in glass reaction tubes at 100 °C for 60 min and chilled in ice. The reaction products were extracted with 5 mL of toluene (Guangzhou Chemistry, Guangzhou, China) and vortexed for 30 s. The color variations were then measured using a spectrophotometer at 520 nm at 25 °C with toluene as a blank control. To measure proline contents in leaves and roots, a calibration curve based on a proline standard was constructed.

### 4.8. The Enzyme Activities and SPAD Values Measurement

Root or leaf tissues (0.1 g each) were fully ground and mixed with 10 mL pre-cooling phosphate buffer (pH 7.8, 0.05 M), respectively, and then centrifuged at 4 °C and 10,000 rpm for 20 min. The supernatant liquid was utilized instantly to assess the enzyme activity. The superoxide dismutase (SOD) activity test was conducted using the previously reported [70]. The reaction system included 0.5 mL plant extracts, 1 mL 125 mM sodium carbonate (Solarbio, Beijing, China), 0.4 mL 25 μM nitro blue tetrazolium (NBT) (Rhawn Chemistry, China), and 0.2 mL 0.1 mM ethylene diamine tetraacetic acid (EDTA) (Guangzhou Chemistry, China). Subsequently, 0.4 mL 1 mM hydroxylamine hydrochloride (Rhawn Chemistry, China) was added to start the reaction, and the absorbance was measured at 560 nm. SOD units were indicated by the quantity of enzyme needed for preventing a 50% decline in NBT.

The peroxidase (POD) activity was measured using the method’s instructions with a minor modification [71]. In a nutshell, POD was measured in a 3 mL total volume of a combination that contained 30% H_2_O_2_ (Guangzhou Chemistry, China) and 1% guaiacol (Sinopharm Group, Beijing, China). For the reaction, 40 μL of the enzyme extraction solution was added to the mixture. At 470 nm, the changes in absorbance caused by guaiacol oxidation were quantified. POD activity units were indicated by the value of OD 470 nm decreased by 0.01 in 1 min.

The activity of catalase (CAT) was assayed using the technique previously described [72]. The mixed solvent (3 mL in total) included 0.1 mL enzyme extracting solution and 2.9 mL reaction solution made from 30% H_2_O_2_ and phosphate buffer (pH 7.0, 0.15 M). CAT activity was calculated by monitoring the decreased value of the absorption spectrum of H_2_O_2_ at 240 nm.

The activity of ascorbate peroxidase (APX) was detected by monitoring the decreased rates of ascorbate oxidation (2.8 mM^−1^ cm^−1^) in absorbance at 290 nm [73]. The reaction mixed solvent (3 mL) was formed by 0.1 mL extracting solution, 2.6 mL 0.1 mM EDTA, 0.15 mL 20 mM H_2_O_2_ and 0.15 mL 5 mM ascorbate (Ghtech, China). APX activity units were indicated by the quantity of enzymes needed for oxidation 1 μM ascorbate. 

The portable chlorophyll measurement instrument (Konica SPAD-502Plus, Tokyo, Japan) was used to determine the Soil and Plant Analyzer Development (SPAD) values of the relative content of chlorophyll of fully unfolded leaves in each seedling [74]. SPAD values were measured at the upper, lower, left, right, and middle parts of each leaf, and then their average values were obtained. Four biological replicates were performed for each experiment.

### 4.9. Concentrations of Metal Ions in the Leaves and Roots

The leaves and roots of peanut plants were obtained individually at 20 days under different Mn treatments to determine metal ion concentrations. After drying and pulverizing, the root and leaf dry samples were weighed 0.2 g separately and placed in the Teflon digestion tank (Chang Yi KH-15, Beijing, China) with 5 mL of 98% H_2_SO_4_ (Kermel, China) and soaked overnight. The digesting tanks were then placed in a drying oven with constant temperature (Yiheng BGP9050AH, Beijing, China), where the temperature was held at 80 °C for 2 h, 120 °C for 2 h, and 160 °C for 4 h. When the samples were in a clear or colorless solution, the digestion was finished. The inside jars and lids of the digesting tanks were washed 3 times with 1% H_2_SO_4_ after the samples were cooled to room temperature. The washing liquor was then transferred into a 50 mL volumetric flask (Robender, Nanchang, China), and the 1% H_2_SO_4_ solution was then replenished to the scale line. The Mn, Fe, and Mg levels in the samples were assessed using the ICP-AES (inductively coupled plasma atomic emission spectrometry; Hitachi PS7800, Tokyo, Japan) [75], with the blank digestion solution serving as a reference. Each index was run 4 times. The kind of metal element was identified using the distinctive spectral wavelengths of the element, and quantitative analysis of the elemental content was carried out by contrasting the strength of the mass spectrometric signal with the concentration of the element.

### 4.10. Preparation of cDNA Libraries and Transcriptomic Sequencing Analysis

The seedlings of peanut were cultivated in hydroponic nutrient solutions added with 10 μM (control group) or 300 μM (Mn toxic stress treatment group) MnSO_4_ as described above. After 20 days, root and leaf samples were respectively obtained for extraction of total RNA and formation of mRNA library, and transcriptomic sequencing analysis was performed according to a previous report [76]. Three biological repetitions were employed for each sample. For RNA extraction, the TRIzol reagent (Invitrogen, Waltham, MA, USA) was used. The samples of RNA were concentrated using ferrite beads containing dT (oligo), followed by fragmentation and reverse-transcription using randomized primers. After being purified, cDNA was handled with terminal modification. Meanwhile, the entire library was generated via polymerase chain reaction (PCR) amplification. The library made sequence determinations by adopting the Illumina platform and the method of PE150 sequencing. Data quality was assured by filtering the raw sequencing results to generate high-class sequences (clean data). The clean data were then aligned with the reference genome of peanut (*Arachis hypogaea PGR*) by application of the system version of TopHat 2.0.12 [77,78].

For the quantification of gene expression levels, HTSeq version 0.6.0 and transcript fragments per kilobase per million reads were employed. The DEGs (differentially expressed genes) were determined using DESeq version 1.16 and corrected with q ≤ 0.05 and |log_2_ ratio| ≥ 1 [79]. Those data were added to Comprehensive Gene Expression Database with the accession number PRJNA901194. DAVID was used to conduct a Gene Ontology (GO) functional enrichment study [80,81]. 

### 4.11. Real-Time Fluorescence Quantitative PCR (qRT-PCR) Testing

The total RNA of peanut plants was isolated from leaves and roots separately by RNA extraction kits (Yeasen, Shanghai, China). After eliminating the gDNA (genomic DNA), cDNA was synthesized using the PrimeScript RT reagent kits (Takara, Maebashi, Japan). For qRT-PCR analysis, the real-time fluorescence quantitative PCR instrument (Bio-Rad, Hercules, CA, USA) was adopted as previously reported [82]. In brief, the cDNA samples were made thinning for 30 folds to become the templates for qRT-PCR testing, and the system of reaction was firstly at 95 °C for 30 s, subsequently followed by 40 cycles at 95 °C keeping for 5 s and at 60 °C keeping for 15 s, followed by 72 °C keeping for 30 s. As a control, the internal control gene *AhUbiquitin* (DQ887087.1) was employed, and the relative transcript level was computed according to the transcript ratios from the selected genes to those from internal control genes, just like previously reported [83]. The primers used for qRT-PCR testing are listed in Appendix A.

### 4.12. Statistic Evaluation

Microsoft Excel 2010 (Excel 14 for Windows) (Microsoft Corporation, Redmond, WA, USA) was used to evaluate all the data. Student’s *t*-test and Duncan multiple comparisons were used for comparison and difference significance analysis.

## 5. Conclusions

The influences of Mn poisoning on peanut leaves and roots were investigated, and the research results indicated that more Mn supplies were delivered to roots than to peanut leaves. In addition, the whole-genome analysis of transcriptome sequencing was performed to distinguish DEGs across the peanut leaves and roots in responding to Mn poisoning. In peanut roots and leaves, 749 and 4589 DEGs were found, respectively. Only 310 DEGs were typically adjusted in both leaves and roots. Our findings add to our understanding of the different reactions of roots and leaves to Mn toxicity.

## Figures and Tables

**Figure 1 ijms-24-01161-f001:**
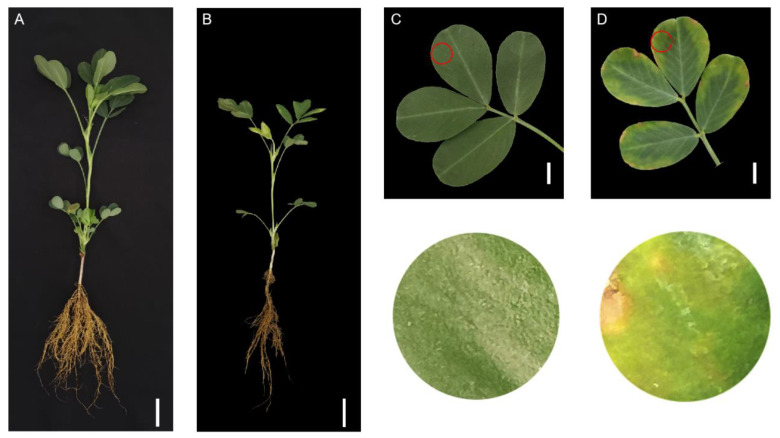
Varying Mn concentrations had different effects on peanut development. Peanut plant morphologies at different Mn concentrations: (**A**) 10 μM, (**B**) 300 μM (bars = 3 cm); leaf phenotypes at different Mn concentrations: (**C**) 10 μM, (**D**) 300 μM (bars = 2 cm). The diameter of the red circle was 2 cm. The figures below show the magnified outcome of the red circle.

**Figure 2 ijms-24-01161-f002:**
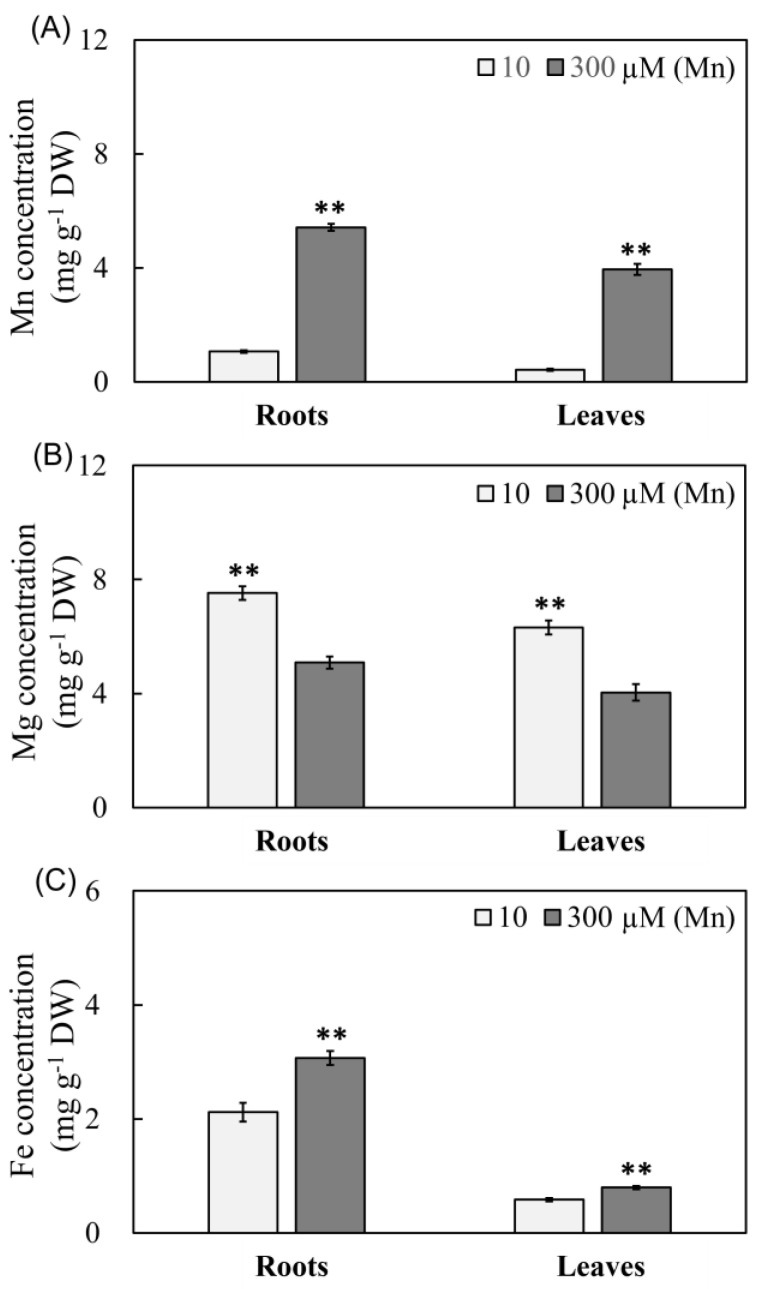
Results of varying Mn treatment concentrations on the Mn, Mg and Fe levels of peanut roots and leaves. (**A**) Mn content; (**B**) Mg content; and (**C**) Fe content. Data was represented via average value and standard deviation of four times experimental replications. Student’s *t*-test was used to assess the significance of the difference between the control and Mn toxicity (** *p* < 0.01).

**Figure 3 ijms-24-01161-f003:**
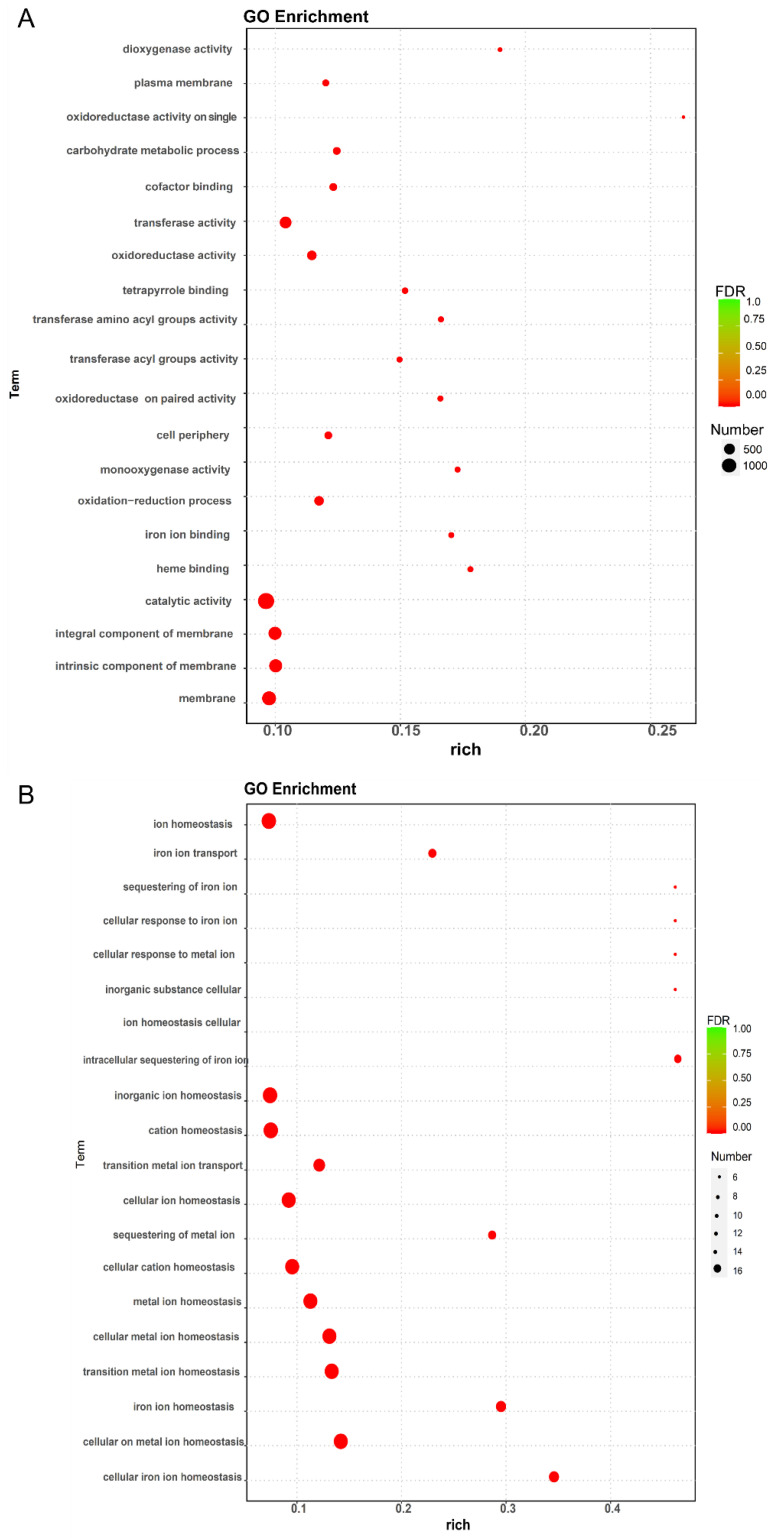
Bubble diagram for GO enrichment analysis. The value of FDR generally ranged from 0-1, and the closer to zero, the more significant the enrichment. The top 20 GO Term entries with the smallest FDR values, i.e., the most significant enrichment, were selected for display. (**A**) CK leaves and 300 µM Mn leaves; (**B**) CK roots and 300 µM Mn roots.3.7. Analysis of DEGs That Act as Conduits and Transporters.

**Figure 4 ijms-24-01161-f004:**
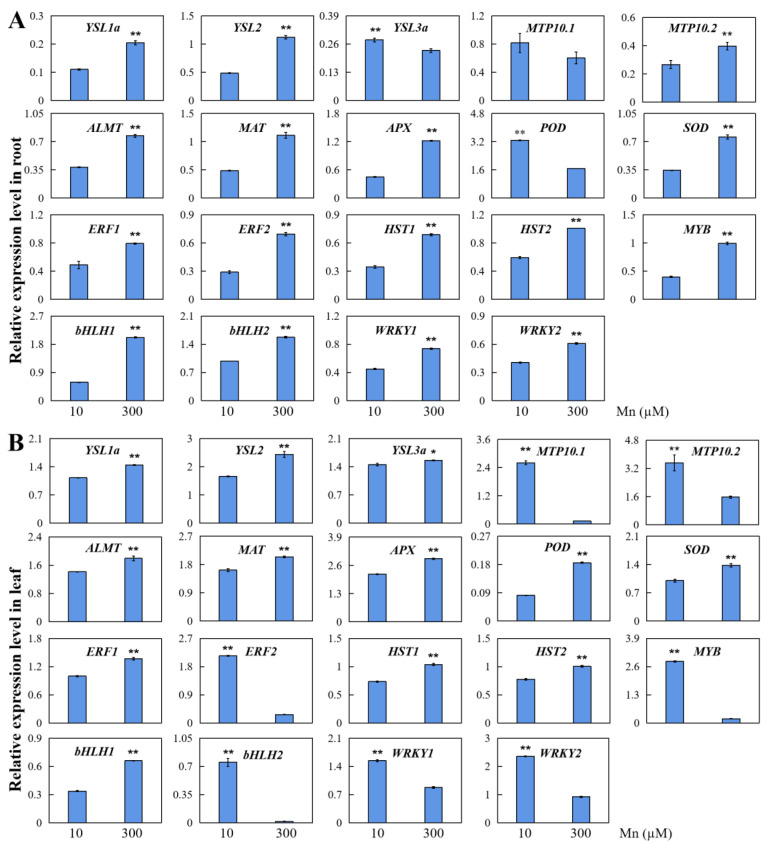
Results of qRT-PCR testing of 19 DEGs in peanut leaves and roots at the concentrations of 10 and 300 μM Mn. The levels of relative expression of DEGs in (**A**) roots and (**B**) leaves. Data was represented via average value and standard deviation of four times experimental replications. Student’s *t*-test was used to assess the significance of the difference between the control and Mn toxicity (* *p* < 0.05, ** *p* < 0.01). YSL: yellow streak protein; MTP: metal resistance protein; ALMT: aluminum activated malate transporter protein; MAT: magnesium transporter protein; SOD: superoxide dismutase; POD: peroxidase; APX: ascorbate peroxidase; ERF: ethylene response transcription factor; HST: transcription factor of heat stress; MYB: transcription factor of MYB; bHLH: transcription factor of bHLH; WRKY: transcription factor of WRKY.

**Figure 5 ijms-24-01161-f005:**
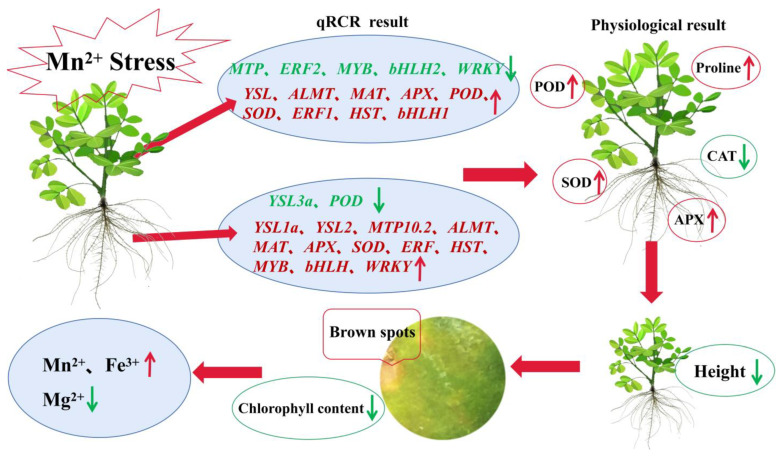
Regulatory processes of peanut roots and leaves in response to Mn toxicity stress. The small red arrows in the graph indicate up-regulation of gene expression or increasing in enzyme activity or substance content, while the small green arrows indicate down-regulation of gene expression or decreasing in enzyme activity or substance content.

**Table 1 ijms-24-01161-t001:** Effects of different concentrations of Mn on peanut development.

Parameters of Peanut Development	Concentrations of Mn (μM)	
	10	300
Height of plant (cm)	25.10 ± 0.36 **	20.93 ± 0.17
SPAD values	44.27 ± 0.09 **	39.53 ± 0.29
Number of brown spots on the fifth leaves	0	60.00 ± 2.45 **
Fresh weights of the shoot (g)	6.06 ± 0.96 **	2.19 ± 0.12
Fresh weights of the root (g)	1.57 ± 0.35 **	0.68 ± 0.02
Shoot dry weights	0.80 ± 0.13 **	0.37 ± 0.03
Root dry weights	0.16 ± 0.03 *	0.08 ± 0.00

Notes: Data was represented via average value and standard deviation of four times experimental replications. Student’s *t*-test was used to assess the significance of the difference between the control and Mn toxicity (* *p* < 0.05, ** *p* < 0.01).

**Table 2 ijms-24-01161-t002:** Effects of different manganese concentrations on peanut root growth.

Parameters of Peanut Root Growth	Concentrations of Mn (μM)	
	10	300
Average diameter of root (mm)	0.89 ± 0.03	0.88 ± 0.01
Volume of root (cm^3^)	184.98 ± 4.88 **	16.71 ± 3.39
Surface area of root (cm^2^)	1393.15 ± 18.23 **	459.97 ± 59.43
Total length of root (cm)	4239.83 ± 730.85 **	1741.37 ± 203.57
Root tip number	12,789.67 ± 781.01 **	6473.67 ± 504.61

Notes: Data was represented via average value and standard deviation of four times experimental replications. Student’s *t*-test was used to assess the significance of the difference between the control and Mn toxicity (** *p* < 0.01).

**Table 3 ijms-24-01161-t003:** Effects of different manganese treatment concentrations on physiological indices in peanut leaves and roots.

Physiological Indices	Leaves		Roots	
	10 μM	300 μM	10 μM	300 μM
Activity of POD (U/g FW)	1833.33 ± 155.90 **	16,500.00 ± 810.09	32,500.00 ± 1503.47	32,000.00 ± 2215.01
Activity of CAT (U/g FW)	1787.33 ± 84.98 **	1516.67 ± 62.36	500.00 ± 40.82 **	133.33 ± 23.57
Activity of APX (U/g FW)	206.67 ± 30.91	603.33 ± 26.25 **	206.67 ± 4.71	376.67 ± 12.47 **
Activity of SOD (U/g FW)	542.36 ± 15.00	832.08 ± 11.60 **	319.80 ± 11.60	388.97 ± 6.18 **
Content of soluble protein (mg/g FW)	36,389.47 ± 561.00 *	33,638.26 ± 816.76	15,787.37 ± 843.39	17,988.34 ± 154.62 *
Content of MDA (μM/g FW)	0.033 ± 0.002 *	0.026 ± 0.002	0.012 ± 0.001	0.014 ± 0.002
Content of proline (μg/g FW)	66.68 ± 17.02	277.55 ± 32.05 **	19.24 ± 2.00	39.38 ± 6.08 **

Notes: Data was represented via average value and standard deviation of four times experimental replications. Student’s *t*-test was used to assess the significance of the difference between the control and Mn toxicity (* *p* < 0.05, ** *p* < 0.01).

**Table 4 ijms-24-01161-t004:** A summary of the analysis of the transcriptome in leaves and roots.

	Number of Total Expressed Genes	Number of Up-Regulated DEGs	Number of Down-Regulated DEGs	Number of DEGs Identified in Both Roots and Leaves	Number of DEGs Identified Only in Roots or Leaves
Leaves	47,675	3566	1023	310	4718
Roots	52,320	542	207		

Note: DEGs represent differential expression genes at two Mn levels.

**Table 5 ijms-24-01161-t005:** Identification of DEGs that act as conduits and transporters.

Gene ID	log_2_FoldChange		Description
	Roots	Leaves	
AH11G27280	1.02	-	Oligopeptide transporter
AH10G04950	1.08	-	Oligopeptide transporter
AH14G01590	−1.97	1.22	Oligopeptide transporter
AH17G01740	-	−1.57	Oligopeptide transporter
AH11G23790	-	−2.13	Oligopeptide transporter
AH01G08540	-	1.46	Oligopeptide transporter
AH11G00400	-	1.02	Oligopeptide transporter
AH14G23400	1.78	1.71	Calcium-transporting ATPase
AH20G34940	1.42	4.30	Calcium-transporting ATPase
AH06G11590	1.27	2.15	Calcium-transporting ATPase
AH16G01740	1.19	1.67	Calcium-transporting ATPase
AH17G01280	1.14	1.71	Calcium-transporting ATPase
AH01G07000	1.04	1.32	Calcium-transporting ATPase
AH12G04160	-	4.24	Calcium-transporting ATPase
AH10G27230	-	3.18	Calcium-transporting ATPase
AH10G27240	-	3.13	Calcium-transporting ATPase
AH07G01100	-	2.68	Calcium-transporting ATPase
AH13G12150	-	2.47	Calcium-transporting ATPase
AH11G01720	-	2.31	Calcium-transporting ATPase
AH19G39830	-	2.04	Calcium-transporting ATPase
AH02G03820	-	1.66	Calcium-transporting ATPase
AH01G06980	-	1.60	Calcium-transporting ATPase
AH15G14410	-	1.53	Calcium-transporting ATPase
AH10G27570	-	1.11	Calcium-transporting ATPase
AH09G00230	-	1.02	Calcium-transporting ATPase
AH15G14430	-	−2.20	Calcium-transporting ATPase
AH15G32580	−1.14	1.24	Metal-nicotianamine transporter YSL
AH09G18140	−3.51	-	Metal-nicotianamine transporter YSL
AH17G27180	-	1.78	Metal-nicotianamine transporter YSL
AH08G03900	-	1.57	Metal-nicotianamine transporter YSL
AH11G01940	-	1.42	Metal-nicotianamine transporter YSL
AH05G22970	-	1.04	Metal-nicotianamine transporter YSL
AH03G40290	2.27	4.22	High-affinity nitrate transporter
AH01G16560	−2.59	-	High-affinity nitrate transporter
AH13G43220	-	1.63	High-affinity nitrate transporter
AH06G25880	-	1.25	High-affinity nitrate transporter
AH03G23240	-	−3.41	High-affinity nitrate transporter
AH10G30290	2.74	-	Metal tolerance protein
AH13G56980	2.53	-	Metal tolerance protein
AH16G14430	-	1.40	Metal tolerance protein
AH09G00220	-	1.10	Metal tolerance protein
AH13G46720	1.88	-	Potassium transporter
AH16G06470	1.52	2.00	Potassium transporter
AH06G03660	1.27	2.96	Potassium transporter
AH01G04720	1.24	-	Potassium transporter
AH11G03840	−1.99	-	Potassium transporter
AH06G07520	-	4.56	Potassium transporter
AH16G14880	-	4.07	Potassium transporter
AH03G40840	-	3.23	Potassium transporter
AH10G19740	-	2.72	Potassium transporter
AH13G43650	-	1.84	Potassium transporter
AH20G23230	-	1.42	Potassium transporter
AH13G50440	-	1.20	Potassium transporter
AH18G28880	1.07	-	Sulfate transporter
AH20G08820	−1.11	4.28	Sulfate transporter
AH10G05870	−1.20	-	Sulfate transporter
AH20G08800	−3.04	-	Sulfate transporter
AH06G11450	-	2.26	Sulfate transporter
AH10G09900	-	1.30	Sulfate transporter
AH14G17090	-	1.01	Sulfate transporter
AH17G34960	-	−1.21	Sulfate transporter
AH06G15690	-	−1.44	Sulfate transporter
AH09G31000	1.66	-	Aluminum-activated malate transporter
AH08G16020	1.37	-	Aluminum-activated malate transporter
AH05G32880	−1.56	−2.16	Aluminum-activated malate transporter
AH13G14340	-	7.25	Aluminum-activated malate transporter
AH06G10950	-	1.60	Aluminum-activated malate transporter
AH16G02290	-	1.26	Aluminum-activated malate transporter
AH05G03780	-	−2.05	Aluminum-activated malate transporter
AH01G13250	-	−2.81	Aluminum-activated malate transporter
AH11G13250	-	−3.12	Aluminum-activated malate transporter
AH05G36140	−1.13	−3.35	Vacuolar iron transporter
AH15G37400	−1.16	−2.89	Vacuolar iron transporter
AH13G48510	−1.58	-	Vacuolar iron transporter
AH03G45850	−1.78	-	Vacuolar iron transporter
AH20G09060	−2.07	-	Vacuolar iron transporter
AH03G45860	−2.18	-	Vacuolar iron transporter
AH19G35130	−2.49	-	Vacuolar iron transporter
AH10G06170	−2.60	-	Vacuolar iron transporter
AH13G48520	−2.67	-	Vacuolar iron transporter
AH13G48530	−3.08	-	Vacuolar iron transporter
AH04G24840	-	−1.70	Vacuolar iron transporter
AH05G30820	1.05	−2.03	Zinc transporter
AH13G05680	−1.45	-	Zinc transporter
AH04G00880	-	3.97	Zinc transporter
AH18G03990	-	1.19	Zinc transporter
AH17G00620	-	−2.31	Zinc transporter
AH15G22860	-	−2.77	Zinc transporter
AH18G31650	−1.39	-	Boron transporter
AH08G26430	−1.40	3.33	Boron transporter
AH12G28290	-	2.05	Magnesium transporter
AH13G08910	-	1.21	Magnesium transporter
AH16G38110	-	1.13	Magnesium transporter
AH16G35450	-	−1.22	Magnesium transporter
AH17G04910	-	3.91	Calcium permeable stress-gated cation channel
AH00G05400	-	3.42	Calcium permeable stress-gated cation channel
AH11G27870	-	2.62	Calcium permeable stress-gated cation channel

Note, “-” represents no difference in gene expression between control and Mn toxicity treatments.

**Table 6 ijms-24-01161-t006:** Identifying DEGs that act as antioxidants.

Gene ID	log_2_FoldChange		Description
	Root	Leaf	
AH13G38160	-	1.28	L-ascorbate oxidase
AH17G04410	-	2.7	L-ascorbate oxidase
AH03G34360	-	3.01	L-ascorbate oxidase
AH07G17320	3.5	-	Peroxidase
AH08G26960	−1.35	-	Peroxidase
AH00G03280	-	2.84	Peroxidase
AH01G05760	2.43	-	Peroxidase
AH01G05780	1.43	-	Peroxidase
AH20G08730	1.32	-	Peroxidase
AH14G25410	1.07	3.29	Peroxidase
AH14G25430	−1.04	-	Peroxidase
AH09G31660	−2.22	−1.15	Peroxidase
AH19G36370	−2.53	-	Peroxidase
AH07G12590	-	5.9	Peroxidase
AH04G21700	-	3.7	Peroxidase
AH20G08720	-	3.36	Peroxidase
AH10G05800	-	2.75	Peroxidase
AH10G05810	-	2.68	Peroxidase
AH04G21680	-	1.15	Peroxidase
AH13G48270	-	−1.78	Peroxidase
AH15G09760	1.5	-	Peroxidase
AH05G12980	1.3	-	Peroxidase
AH09G08990	1.21	-	Peroxidase
AH15G33990	1.14	-	Peroxidase
AH20G14810	1.09	-	Peroxidase
AH18G05400	1.02	−2.08	Peroxidase
AH08G15110	−1.07	-	Peroxidase
AH04G09840	−1.07	1.53	Peroxidase
AH14G08440	−1.1	1.95	Peroxidase
AH01G05770	−1.19	-	Peroxidase
AH10G20050	−1.29	-	Peroxidase
AH06G20840	−1.99	-	Peroxidase
AH16G25800	−2.23	-	Peroxidase
AH14G08450	-	6.33	Peroxidase
AH14G08420	-	5.79	Peroxidase
AH18G10570	-	4.14	Peroxidase
AH06G26990	-	3.94	Peroxidase
AH04G09830	-	3.37	Peroxidase
AH04G09870	-	2.71	Peroxidase
AH17G11990	-	2.29	Peroxidase
AH14G08430	-	1.96	Peroxidase
AH06G24710	-	1.91	Peroxidase
AH01G31200	-	1.87	Peroxidase
AH12G38300	-	1.63	Peroxidase
AH04G09850	-	1.63	Peroxidase
AH16G25780	-	1.62	Peroxidase
AH09G19280	-	1.48	Peroxidase
AH15G01790	-	1.1	Peroxidase
AH11G28810	-	−1.05	Peroxidase
AH01G21450	-	−1.15	Peroxidase
AH10G10440	-	−1.32	Peroxidase
AH00G04650	-	−1.79	Peroxidase
AH18G07180	-	−2.49	Peroxidase
AH03G05320	1.22	-	Peroxidase
AH03G07350	-	3.32	Peroxidase
AH13G09650	-	2.47	Peroxidase
AH19G19840	-	2.6	Superoxide dismutase

Note, “-” represents no difference in gene expression between control and Mn toxicity treatments.

**Table 7 ijms-24-01161-t007:** Finding of DEGs that serve as transcription factors.

Gene ID	log_2_FoldChange		Description
	Root	Leaf	
AH19G30930	-	1.24	AP2/ERF and B3 domain-containing transcription factor
AH09G30280	-	1.03	AP2/ERF and B3 domain-containing transcription factor
AH17G24920	-	1.26	bZIP transcription factor
AH02G13610	-	1.22	bZIP transcription factor
AH02G08810	-	1.82	Ethylene-responsive transcription factor
AH03G24670	-	5.29	Ethylene-responsive transcription factor
AH19G42640	-	6.77	Ethylene-responsive transcription factor
AH18G34060	-	4.82	Ethylene-responsive transcription factor
AH13G35500	-	2.19	Ethylene-responsive transcription factor
AH02G05220	-	−1.23	Ethylene-responsive transcription factor
AH03G39850	-	3.87	Ethylene-responsive transcription factor
AH16G07170	-	3.34	Ethylene-responsive transcription factor
AH09G10470	1.22	-	Ethylene-responsive transcription factor
AH17G30770	−1.38	-	Ethylene-responsive transcription factor
AH20G26610	1.2	2.71	Ethylene-responsive transcription factor
AH16G37720	1.03	1.09	Ethylene-responsive transcription factor
AH20G06980	-	1.35	Ethylene-responsive transcription factor
AH05G03100	-	1.39	Ethylene-responsive transcription factor
AH09G21370	-	1.77	Ethylene-responsive transcription factor
AH08G21930	-	2.49	Ethylene-responsive transcription factor
AH10G02570	-	2.18	Ethylene-responsive transcription factor
AH20G07000	-	2.07	Ethylene-responsive transcription factor
AH11G14950	-	2.1	Ethylene-responsive transcription factor
AH05G03090	-	1.97	Ethylene-responsive transcription factor
AH01G14550	-	1.17	Ethylene-responsive transcription factor
AH03G15340	-	3.27	Ethylene-responsive transcription factor
AH13G17810	-	3.2	Ethylene-responsive transcription factor
AH10G20310	-	3.88	Ethylene-responsive transcription factor
AH13G27750	-	4.36	Ethylene-responsive transcription factor
AH16G13670	-	−1.67	Ethylene-responsive transcription factor
AH09G25330	-	2.02	Ethylene-responsive transcription factor
AH08G17790	-	2.72	Ethylene-responsive transcription factor
AH13G32380	-	2.07	Ethylene-responsive transcription factor
AH12G16570	-	3.32	Ethylene-responsive transcription factor
AH02G14060	-	2.8	Ethylene-responsive transcription factor
AH10G14340	-	5.22	Ethylene-responsive transcription factor
AH20G18990	-	5.12	Ethylene-responsive transcription factor
AH10G01270	-	1.11	Ethylene-responsive transcription factor
AH06G04120	-	6.22	Ethylene-responsive transcription factor
AH17G18750	-	2.06	Ethylene-responsive transcription factor
AH20G13550	-	1.39	Ethylene-responsive transcription factor
AH03G01980	-	−1.4	Ethylene-responsive transcription factor
AH00G01740	-	−3.06	Ethylene-responsive transcription factor
AH19G32730	-	1.65	GATA transcription factor
AH09G34390	-	1.64	GATA transcription factor
AH01G26750	-	1.38	GATA transcription factor
AH12G17550	-	−1.08	GATA transcription factor
AH01G21520	-	2.2	Heat stress transcription factor
AH06G11540	-	2.31	Heat stress transcription factor
AH13G40070	-	3.18	Heat stress transcription factor
AH03G36820	-	3.21	Heat stress transcription factor
AH15G35830	-	2.7	Heat stress transcription factor
AH05G34690	-	2.26	Heat stress transcription factor
AH05G17170	-	1.21	Heat stress transcription factor
AH06G02170	-	1.31	Heat stress transcription factor
AH05G34050	-	1.27	MADS-box transcription factor
AH01G14370	4.18	-	Myb family transcription factor
AH14G23270	-	−1.95	Myb family transcription factor
AH12G25410	-	−2.45	Myb family transcription factor
AH14G07590	-	1.61	Myb family transcription factor
AH13G01940	-	−1.4	Myb family transcription factor
AH14G23270	−2.11	-	Myb family transcription factor
AH04G20450	-	−2.16	Myb family transcription factor
AH02G22970	-	−1.89	Myb family transcription factor
AH01G14370	-	2	Myb family transcription factor
AH13G54280	-	−1.38	Myb family transcription factor
AH19G15410	−4.5	-	Myb family transcription factor
AH04G18490	-	−2.38	Myb family transcription factor
AH06G04320	-	−2.25	Myb family transcription factor
AH03G21480	-	1.72	Myb family transcription factor
AH08G29090	-	4.62	Myb family transcription factor
AH12G03980	-	3.02	Myb family transcription factor
AH14G44850	-	−1.8	Myb family transcription factor
AH16G39650	-	1.59	Myb family transcription factor
AH08G03280	-	3.94	Myb family transcription factor
AH17G26480	-	5.78	Myb family transcription factor
AH16G42280	-	2.5	Myb family transcription factor
AH03G42580	-	1.16	NAC transcription factor
AH08G11600	-	−1.73	Nuclear transcription factor
AH05G27340	-	1.32	BEE transcription factor
AH03G21460	-	1.39	BEE transcription factor
AH08G08420	−3.78	-	bHLH transcription factor
AH16G10910	−1.12	-	bHLH transcription factor
AH09G15020	−1.67	-	bHLH transcription factor
AH03G03430	-	−1.13	bHLH transcription factor
AH08G25020	-	1.32	bHLH transcription factor
AH18G29680	-	1.68	bHLH transcription factor
AH11G35750	-	5.31	bHLH transcription factor
AH01G22060	-	4.16	bHLH transcription factor
AH03G05080	-	1.58	bHLH transcription factor
AH02G04370	-	−2.12	bHLH transcription factor
AH12G04850	-	−1.99	bHLH transcription factor
AH05G39020	-	1.01	bHLH transcription factor
AH18G02090	-	−1.03	bHLH transcription factor
AH17G15960	-	−2.33	bHLH transcription factor
AH07G20060	-	1.29	bHLH transcription factor
AH17G17690	-	1.36	bHLH transcription factor
AH05G16400	-	5.21	CPC transcription factor
AH15G06520	-	5.2	CPC transcription factor
AH17G07910	-	1.2	DIVARICATA transcription factor
AH09G29640	-	1.06	DIVARICATA transcription factor
AH15G18860	-	−1.41	FAMA transcription factor
AH16G03850	-	3.04	HBP-1b transcription factor
AH02G16860	-	1.68	HBP-1b transcription factor
AH12G19950	-	4.38	HBP-1b transcription factor
AH17G10210	−2.85	-	KAN transcription factor
AH16G05200	-	1.45	ORG transcription factor
AH19G38650	-	1.85	PERIANTHIA transcription factor
AH20G33360	1.28	-	TCP transcription factor
AH14G42340	-	6.33	TCP transcription factor
AH19G37380	-	1.11	TCP transcription factor
AH03G34200	-	−1.01	TGA transcription factor
AH20G00510	-	−1.45	UNE transcription factor
AH08G30400	-	−1.8	UNE transcription factor
AH14G41690	3.89	-	Trihelix transcription factor
AH12G24650	-	−2.2	Trihelix transcription factor
AH09G16910	-	−1.56	Trihelix transcription factor
AH07G03020	1.44	-	WRKY transcription factor
AH01G21930	-	1.58	WRKY transcription factor
AH13G34700	-	5.27	WRKY transcription factor
AH08G28680	1.86	2.74	WRKY transcription factor
AH12G03520	2.21	2.73	WRKY transcription factor
AH16G13340	2.05	4.06	WRKY transcription factor
AH06G25830	1.1	2.83	WRKY transcription factor
AH03G28760	1.57	3.18	WRKY transcription factor
AH13G32420	1.44	2.71	WRKY transcription factor
AH08G09100	1.4	2.93	WRKY transcription factor
AH03G21920	-	1.23	WRKY transcription factor
AH13G25080	-	1.18	WRKY transcription factor
AH08G25500	-	2.38	WRKY transcription factor
AH18G30430	-	2	WRKY transcription factor
AH06G09380	-	3.71	WRKY transcription factor
AH16G32150	-	2.65	WRKY transcription factor
AH19G01540	-	2.17	WRKY transcription factor
AH09G00690	-	2.31	WRKY transcription factor
AH17G33510	-	2.27	WRKY transcription factor
AH13G37760	-	4.33	WRKY transcription factor
AH03G33940	-	4.02	WRKY transcription factor
AH08G19200	-	1.47	WRKY transcription factor
AH10G17080	-	3.06	WRKY transcription factor
AH20G22960	-	2.1	WRKY transcription factor
AH17G03120	-	5.7	WRKY transcription factor
AH06G24570	-	1.53	WRKY transcription factor
AH03G30560	-	3.58	WRKY transcription factor
AH16G30280	-	1.28	WRKY transcription factor
AH01G28930	-	1.48	WRKY transcription factor
AH11G28210	-	1.66	WRKY transcription factor

Note, “-” represents no difference in gene expression between control and Mn toxicity treatments.

## Data Availability

The original contributions presented in the study are publicly available. This RAN-seq raw data can be found on the NCBI repository, accession number: PRJNA901194.

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
