# Peer review of "Comparative Transcriptome Analysis Reveals Complex Physiological Response and Gene Regulation in Peanut Roots and Leaves under Manganese Toxicity Stress"

_ijms, 2023, doi:10.3390/ijms24021161_

Round 1

Reviewer 1 Report

The objective of this study is to test the physiological response and gene regulation of peanut roots and leaves under manganese toxicity stress manuscript – as mentioned in the title.  

 In my opinion, the experimental design is not a proper one (at least two more concentrations of Mn would have been appropriate in the range of the tested 10 and 300 uM), hence the findings reported here are quite limited and not enough to be published in a journal such as the International Journal of Molecular Sciences.

 However, if the editor’s decision will be to keep it, please consider that it needs a revision; here are some suggestions for an improved version:

 #Abstract – add a phrase clearly stating the goal of this study

L.12, 35 – delete “(Mn)” – common symbol

L. 51 – use capital in “christoffers

L. 57 – delete the acronym “(IAA)” – not used later

L. 61 – replace “chain of food” > food chain

#1 – clearly define the objective of this study

#2 – add a new paragraph – give relevant details about the chemical substances used

L.128 – 129 – how was the pH adjusted?

L.130 – what does it means “different concentrations”? Give a proper experimental design for this study

L.160 – delete “Catalase” – the acronym was already defined

#2.5 and #2.6 – give relevant information on each of the procedure used, not only the references: sample amounts, sample preparation techniques, types and producers for the equipment and reagents used, wavelengths used for spectrophotometric methods, as well as for the accomplished calibrations

L. 215 – delete “The software of

L. 216 - use capital in “duncan

L.235 – 238 – if this text represents notes for figure 1, consider using smaller letters and a proper indentation, otherwise rephrase it.

#Figure 2, #Figure 3, #Figure 4, figure 7 - histograms with only two bars looks really bad for a scientific paper; consider converting these figure into tables

#Table 4 – improper format

L.521, 526 - replace “photo synthesis” > photosynthesis

L.556, 558, 560, 614 – rephrase “vitalities“ – inaccurate technical language

#References – check the alphabetical order and make appropriate corrections

L.844 – 845 – silicon is outside the study area, as well as the plant material used there, hence consider deleting this reference

Author Response

  1. #Abstract – add a phrase clearly stating the goal of this study

My response: The findings of this study can serve as a starting point for further research into the molecular mechanism of important functional genes in peanut roots and leaves that regulate peanut tolerance to manganese poisoning.

  1. L.12, 35 – delete “(Mn)” – common symbol

My response: L.12, 35 – the common symbols of “(Mn)” have been deleted.

  1. L. 51 – use capital in “christoffers”

My response: Line 233 “christoffers” has been changed into “Christoffers ".

  1. L. 57 – delete the acronym “(IAA)” – not used later

My response: L. 57 “(IAA)” has been deleted.

  1. L. 61 – replace “chain of food” > food chain

My response: L. 61 – “chain of food” has been replaced with “food chain”.

  1. #1 – clearly define the objective of this study

My response: The results of this study may provide a preliminary basis for additional research into the specific functions of genes that are sensitive to Mn toxicity.

  1. #2 – add a new paragraph – give relevant details about the chemical substances used

My response: All chemicals used are analytically pure grade reagents (Kermel, China).

  1. L.128 – 129 – how was the pH adjusted?

My response: And every two days the pH of the nutrient solution was adjusted to 5.0 with 1 mol/L of KOH or H2SO4 (Kermel, China).

  1. L.130 – what does it means “different concentrations”? Give a proper experimental design for this study

My response: After 20 days of treatment with 10 and 300 mM of Mn, the above-ground and below-ground parts of peanuts were collected individually to determine the fresh and dry weights, the roots and number of manganese spots on the fifth trifoliate leaves, etc.

  1. L.160 – delete “Catalase” – the acronym was already defined

My response: OK. We have deleted the “Catalase”.

  1. #2.5 and #2.6 – give relevant information on each of the procedure used, not only the references: sample amounts, sample preparation techniques, types and producers for the equipment and reagents used, wavelengths used for spectrophotometric methods, as well as for the accomplished calibrations

My response: That's an excellent suggestion. We have revised the manuscript as required as the suggestions.

2.5. Malondialdehyde (MDA) content assaying

The physiological response indicators of peanut roots and leaves were examined after 20 days of dealing with 10 μM (control group, normal Mn concentration) and 300 μM (Mn toxicity stress concentration) MnSO4. The content of MDA was determined by using a slightly modified version of thiobarbituric acid (TBA) technique [35]. In brief, 0.1 g leaf or root tissue were homogenized in 10 mL phosphate buffer (Solarbio, China) (pH 7.8, 0.05 M) before being extracted in 2 mL 0.6% TBA (Rhawn, China). The extract was place in the thermostatic water bath (Shanghai Lichen, China) at 100 °C for 15 minutes before being rapidly cooled on ice. After spinning at 4,000 rpm for 20 minutes in a centrifugal machine (Eppendorf 5415D, Germany), the absorbance of the liquid supernatant was tested at 450, 532, and 600 nm respectively using an ultraviolet-visible spectrophotometer (Shanghai Yuanxi UV-5100B, China) with a 10 mm quartz cuvette (Allrenta, China). The compound of MDA-TBA was quantized by the extinction coefficients (155 mM-1 cm-1).

2.6. Soluble protein content assaying

The coomassie brilliant blue technique was used to determine the amount of soluble protein [36]. 0.1 g leaf or root tissue were homogenized respectively in 10 mL phosphate buffer (pH 7.8, 0.05 M) before being extracted in 2.9 mL liquid mixture containing 0.1 g Coomassie brilliant blue G-250 (Rhawn Chemistry, China). After a 2-minute reaction, the absorbance of the liquid supernatant was tested at 595 nm to calculate the protein concentration in the sample by using a standard curve via bovine serum albumin (BSA) (Rhawn Chemistry, China).

2.7. Proline content analysis

Leaf and root samples (0.1 g each) were mixed in 10 mL 3% sulfosalicylic acid (Rhawn Chemistry, China) before filtering to evaluate the proline concentrations of the leaves and roots, respectively [37]. The reaction mixture contained 2 mL extracted supernatant, 3 mL acid ninhydrin reagent (Rhawn Chemistry, China) and 2 mL glacial acetic acid (Ghtech, China) was place in glass reaction tubes at 100 °C for 60 minutes and chilled in ice. The reaction products were extracted with 5 mL of toluene (Guangzhou Chemistry, China) and vortexed for 30 s. The color variations were then measured using a spectrophotometer at 520 nm at 25℃ with toluene as a blank control. To measure proline contents in leaves and roots, a calibration curve based on proline standard was constructed.

2.8. The enzyme activities and SPAD values measurement

Root and leaf tissues (0.1 g each) were fully grinded and mixed with 10 mL pre-cooling phosphate buffer (pH 7.8, 0.05 M), respectively, and then centrifuged at 4 °C and 10,000 rpm for 20 minutes. The supernatant liquid was utilized instantly to assess the enzyme activity. The superoxide dismutase (SOD) activity test was conducted using the previously reported [38]. The reaction system included 0.5 mL plant extracts, 1 mL 125 mM sodium carbonate (Solarbio, China), 0.4 mL 25 μM nitro blue tetrazolium (NBT) (Rhawn Chemistry, China), and 0.2 mL 0.1 mM ethylene diamine tetraacetic acid (EDTA) (Guangzhou Chemistry, China). Subsequently, 0.4 mL 1 mM hydroxylamine hydrochloride (Rhawn Chemistry, China) was added to start the reaction, and the absorbance was measured at 560 nm. SOD units were indicated by quantity of enzyme needed for preventing a 50% decline of NBT.

The peroxidase (POD) activity was measured using the method's instructions with a minor modification [39]. In a nutshell, POD was measured in a 3 mL total volume of a combination that contained 30% H2O2 (Guangzhou Chemistry, China) and 1% guaiacol (Sinopharm Group, China). For the reaction, 40 μL of the enzyme extraction solution were added to the mixture. At 470 nm, the changes of absorbance caused by guaiacol oxidation was quantified. POD activity units were indicated by the value of OD 470 nm decreased by 0.01 in 1 min.

The activity of catalase (CAT) was assayed using the technique previously described [40]. The mixed solvent (3 mL in total) included 0.1 mL enzyme extracting solution and 2.9 mL reaction solution made from 30% H2O2 and phosphate buffer (pH 7.0, 0.15 M). CAT activity was calculated by monitoring the decreased value of absorption spectrum of H2O2 at 240 nm.

The activity ascorbate peroxidase (APX) was detected by monitoring the decreased rates of ascorbate oxidation (2.8 mM-1 cm-1) in absorbance at 290 nm [41]. The reaction mixed solvent (3 mL) was formed by 0.1 mL extracting solution, 2.6 mL 0.1 mM EDTA, 0.15 mL 20 mM H2O2 and 0.15 mL 5 mM ascorbate (Ghtech, China). APX activity units were indicated by the quantity of enzymes needed for oxidation 1 μM ascorbate.

The portable chlorophyll measurement instrument (Konica SPAD-502Plus, Japan) was used to determine the Soil and Plant Analyzer Development (SPAD) values of relative content of chlorophyll of fully unfolded leaves in each seedling [42]. SPAD values were measured at the upper, lower, left, right and middle parts of each leaf, and then their average values were obtained. Four biological replicates were performed for each experiment.

2.9. Concentrations of metal ions in the leaves and roots

The leaves and roots of peanut were obtained individually at 20 days under different Mn treatments to determine metal ion concentrations. After drying and pulverizing, the root and leaf dry samples were weighed 0.2 g separately and placed in the Teflon digestion tank (Chang Yi KH-15, China) with 5 mL 98% H2SO4 (Kermel, China) and soaked for overnight. The digesting tanks were then placed in a drying oven with constant temperature (Yiheng BGP9050AH, China), where the temperature was held at 80°C for 2 hours, 120°C for 2 hours, and 160°C for 4 hours. And when the samples were in a clear or colorless solution, the digestion was finished. The inside jars and lids of the digesting tanks were washed three times with 1% H2SO4 after the samples cooled to room temperature. The washing liquor was then transferred into a 50 mL volumetric flask (Robender, China), and the 1% H2SO4 solution was then replenished to the scale line. The Mn, Fe, and Mg levels in the samples were assessed using the ICP-AES (inductively coupled plasma atomic emission spectrometry) (Hitachi PS7800, Japan) [43], with the blank digestion solution serving as a reference. Each index was made run for four times. The kind of metal element was identified using the distinctive spectral wavelengths of the element, and quantitative analysis of the elemental content was carried out by contrasting the strength of the mass spectrometric signal with the concentration of the element.

  1. L. 215 – delete “The software of”

My response: OK. We have deleted the “The software of”.

  1. L. 216 - use capital in “duncan”

My response: L. 216 “duncan” has been changed into “Duncan".

  1. L.235 – 238 – if this text represents notes for figure 1, consider using smaller letters and a proper indentation, otherwise rephrase it.

My response: We have used smaller letters and a proper indentation in text represents notes for figure 1.

  1. #Figure 2, #Figure 3, #Figure 4, figure 7 - histograms with only two bars looks really bad for a scientific paper; consider converting these figures into tables

#Table 4 – improper format

My response: The Figure 2, Figure 3, Figure 4 have been converted into Table 1, Table 2, Table 3, respectively. The results of fluorescence quantitative PCR (Figure 7) are usually shown in the form of a bar graph. In addition, the format of the #Table 4 has been modified.

Table 1. Effects of different concentrations of Mn on peanut development.

Parameters of peanut development

Concentrations of Mn (μM)

10

300

Height of plant (cm)

25.10±0.36**

20.93±0.17

SPAD values

44.27±0.09**

39.53±0.29

Number of brown spots on the fifth leaves

0

60.00±2.45**

Fresh weights of the shoot (g)

6.06±0.96**

2.19±0.12

Fresh weights of the root (g)

1.57±0.35**

0.68±0.02

Shoot dry weights

0.80±0.13**

0.37±0.03

Root dry weights

0.16±0.03*

0.08±0.00

Notes: Data was represented via average value and standard deviation of four times experimental replications. Student’s t-test was used to assess the significance of difference between the control and Mn toxicity (*p < 0.05, **p < 0.01).

Table 2. Effects of different manganese concentrations on peanut root growth.

Parameters of peanut root growth

Concentrations of Mn (μM)

10

300

Average diameter of root (mm)

0.89±0.03

0.88±0.01

Volume of root (cm3)

184.98±4.88**

16.71±3.39

Surface area of root (cm2)

1393.15±18.23**

459.97±59.43

Total length of root (cm)

4239.83±730.85**

1741.37±203.57

Root tip number

12789.67±781.01**

6473.67±504.61

Notes: Data was represented via average value and standard deviation of four times experimental replications. Student’s t-test was used to assess the significance of difference between the control and Mn toxicity (*p < 0.05, **p < 0.01).

Table 3. Effects of different manganese treatment concentrations on physiological indices in peanut leaves and roots.

Physiological indices

Leaves

Roots

10 μM

300 μM

10 μM

300 μM

Activity of POD (U/g FW)

1833.33±155.90**

16500.00±810.09

32500.00±1503.47

32000.00±2215.01

Activity of CAT (U/g FW)

1787.33±84.98**

1516.67±62.36

500.00±40.82**

133.33±23.57

Activity of APX (U/g FW)

206.67±30.91

603.33±26.25**

206.67±4.71

376.67±12.47**

Activity of SOD (U/g FW)

542.36±15.00

832.08±11.60**

319.80±11.60

388.97±6.18**

Content of soluble protein (mg/g FW)

36389.47±561.00*

33638.26±816.76

15787.37±843.39

17988.34±154.62*

Content of MDA (μM/g FW)

0.033±0.002*

0.026±0.002

0.012±0.001

0.014±0.002

Content of proline (μg/g FW)

66.68±17.02

277.55±32.05**

19.24±2.00

39.38±6.08**

Notes: Data was represented via average value and standard deviation of four times experimental replications. Student’s t-test was used to assess the significance of difference between the control and Mn toxicity (*p < 0.05, **p < 0.01).

  1. L.521, 526 - replace “photo synthesis” > photosynthesis

My response: L.521, 526 “photo synthesis” have been changed into “photosynthesis".

  1. L.556, 558, 560, 614 – rephrase “vitalities“ – inaccurate technical language

My response: L.556, 558, 560, 614 “vitalities” have been changed into “activities".

  1. #References – check the alphabetical order and make appropriate corrections

My response: OK. We have checked the alphabetical order and made appropriate corrections.

  1. L.844 – 845 – silicon is outside the study area, as well as the plant material used there, hence consider deleting this reference

My response: L.844 – 845 OK. We have deleted the reference.

Reviewer 2 Report

Comment 1: Rewrite the title.

Comment 2: There is a lot of information. Please rewrite the abstract, very clearly.

Comment 3: Please use Manganese or Mn for subsequent sentences both in the abstract and body of the text. Please check throughout the manuscript.

Comment 4: There is much information in the results, but recent references are missing in the introduction and discussion sections.

Comment 5: Please check author instructions for page numbers, references, and citations in the main text. Please follow the author's instructions.

Author Response

Comment 1: Rewrite the title.

My response: Comparative transcriptome analysis reveals complex physiological response and gene regulation in peanut roots and leaves under Mn toxicity stress

Comment 2: There is a lot of information. Please rewrite the abstract, very clearly.

My response:

Excess Mn is toxic for plant and reduces crop production. Although physiological and molecular pathways may drive plant responses to Mn toxicity, few studies have evaluated Mn tolerance capacity in roots and leaves. As a result, the processes behind Mn tolerance in various plant tissue or organ are unclear. The reactivity of peanut (Arachis hypogaea) to Mn toxicity stress was examined in this study. Mn oxidation spots developed on peanut leaves, and the root growth was inhibited under Mn toxicity stress. The physiological results revealed that under Mn toxicity stress, the activities of antioxidases and the content of proline in roots and leaves were greatly elevated, whereas the content of soluble protein decreased. In addition, manganese and iron ion content in roots and leaves increased significantly, but magnesium ion content decreased drastically. The differentially expressed genes (DEGs) in peanut roots and leaves in response to Mn toxicity were subsequently identified using a genome-wide transcriptome analysis. Transcriptomic profiling results showed that 731 and 4,589 DEGs were discovered individually in roots and leaves. Furthermore, only 310 DEGs were frequently adjusted and controlled in peanut roots and leaves, indicating peanut roots and leaves exhibited various toxicity responses to Mn. The results of qRT-PCR suggested that the gene expression of many DEGs in roots and leaves was inconsistent, indicating a more complex regulation of DEGs. Therefore, different regulatory mechanisms are present in peanut roots and leaves in response to Mn toxicity stress. The findings of this study can serve as a starting point for further research into the molecular mechanism of important functional genes in peanut roots and leaves that regulate peanut tolerance to Mn poisoning.

Comment 3: Please use Manganese or Mn for subsequent sentences both in the abstract and body of the text. Please check throughout the manuscript.

My response: OK. We have used “Mn” in the abstract and body of the text.

Comment 4: There is much information in the results, but recent references are missing in the introduction and discussion sections.

My response: That's an excellent suggestion. We have added a lot of the latest references in the introduction and discussion sections.

Comment 5: Please check author instructions for page numbers, references, and citations in the main text. Please follow the author's instructions.

My response: OK. We have revised the manuscript as required as the author's instructions.
